# *fhl2b* mediates extraocular muscle protection in zebrafish models of muscular dystrophies and its ectopic expression ameliorates affected body muscles

Nils Dennhag [1,2], Abraha Kahsay [1,2], Itzel Nissen [3,4], Hanna Nord [1], Maria Chermenina[1,2], Jiao Liu[5,6], Anders Arner[5], Jing-Xia Liu [1], Ludvig J. Backman[1], Silvia Remeseiro [3,4], Jonas von Hofsten [1,7] & Fatima Pedrosa Domellöf [1,2,7]

In muscular dystrophies, muscle fibers loose integrity and die, causing significant suffering and premature death. Strikingly, the extraocular muscles (EOMs) are spared, functioning well despite the disease progression. Although EOMs have been shown to differ from body musculature, the mechanisms underlying this inherent resistance to muscle dystrophies remain unknown. Here, we demonstrate important differences in gene expression as a response to muscle dystrophies between the EOMs and trunk muscles in zebrafish via transcriptomic profiling. We show that the LIM-protein Fhl2 is increased in response to the knockout of *desmin*, *plectin* and *obscurin*, cytoskeletal proteins whose knockout causes different muscle dystrophies, and contributes to disease protection of the EOMs. Moreover, we show that ectopic expression of *fhl2b* can partially rescue the muscle phenotype in the zebrafish Duchenne muscular dystrophy model *sapje*, significantly improving their survival. Therefore, Fhl2 is a protective agent and a candidate target gene for therapy of muscular dystrophies.

Muscular dystrophies, caused by mutations in more than 40 genes[1] share similar features including muscle fiber disruption leading to muscle weakness, loss of ambulation, and often premature death predominantly due to respiratory failure. Although muscular dystrophies affect 1:4–5000 births worldwide[2] and induce pronounced suffering, there is currently no cure, and treatment options are sparse. Thus, there is a need for new effective treatment options to prolong and improve the quality of life of patients suffering from these detrimental diseases.

The most common form of muscular dystrophy is Duchenne muscular dystrophy (DMD). In DMD, the giant protein dystrophin is lost or truncated[3,4]. Dystrophin is a crucial member of the dystrophin-glycoprotein-complex (DGC). The DGC links the extracellular matrix, across the cell membrane, to F-actin in the cytoskeleton within the myofibrils, which is fundamental for myofiber integrity[5]. The DGC contributes to multiple functions in the myofiber such as force transmission across the sarcolemma, but also acts as a signaling hub[6–8]. Dystrophin is therefore a key element for myofiber integrity. The most

[1]Department of Medical and Translational Biology, Umeå University, Umeå, Sweden. [2]Department of Clinical Sciences, Ophthalmology, Umeå University, Umeå, Sweden. [3]Department of Medical and Translational Biology; Section of Molecular Medicine, Umeå University, Umeå, Sweden. [4]Wallenberg Center for Molecular Medicine (WCMM), Umeå University, Umeå, Sweden. [5]Div. Thoracic Surgery, Dept. Clinical Sciences, Lund University, Lund, Sweden. [6]College of Life Sciences, South-Central University for Nationalities, Wuhan, China. [7]These authors jointly supervised this work: Jonas von Hofsten, Fatima Pedrosa Domellöf. ✉e-mail: Jonas.von.hofsten@umu.se; fatima.pedrosa-domellof@umu.se

frequently used zebrafish DMD model is the *sapje* line[9,10]. *Sapje* zebrafish carry an A-to-T transversion in exon 4 of the dystrophin gene, resulting in a premature stop codon[9]. The lack of dystrophin in zebrafish subsequently leads to detachment of trunk myofibers from the myosepta and failure of the contractile apparatus which ultimately results in the premature death of up to 50% of zebrafish larvae at the age of 5 days post fertilization (dpf)[11]. Hence, the *sapje* zebrafish are severely affected by the lack of dystrophin, but generally mimic the human DMD condition and constitute a good model for experimental treatment studies.

The EOMs have shown an innate resistance towards muscular dystrophies[12–15]. Even though EOMs share attributes with other striated skeletal muscles they differ in terms of gene expression and protein content[16,17] as well as neuromuscular and myotendinous junction composition[18,19]. We have previously shown that zebrafish EOMs are a good model to study the cytoskeleton as well as myofiber and neuromuscular junction cytoarchitecture[20]. Recently, the muscle-specific intermediate filament protein desmin was shown to be naturally lacking in a subset of EOM myofibers in humans and zebrafish[20,21]. Additionally, other intermediate filament proteins such as nestin and keratin-19 show a complex pattern in human EOM myotendinous junctions[19]. Altogether these findings suggest that the cytoskeletal composition of the EOMs differs from that of other skeletal muscles in the body.

Desmin is the most abundant cytoskeletal protein in skeletal muscle fibers[22] and, besides its interaction with the DGC, desmin has been suggested to contribute to gene regulation[23] and also as a mechanosensor, utilizing mechanical stretch to trigger intracellular signaling[24]. Additionally, desmin anchors myonuclei and mitochondria to the sarcolemma and myofibrils[25], reviewed in[26,27]. Despite this important role in the myofiber, studies of *Des* knockout mice have shown that the EOMs are relatively unaffected[28]. Patients with desminopathy have near normal life expectancy, with minor complications compared to DMD patients[29,30]. Desmin mutant animal models therefore offer a route to study the EOMs in a dystrophic background over extended periods of time. We hypothesized that the transcriptome of the EOMs in a dystrophic setting would provide information regarding their innate resistance towards muscular dystrophies, and that this could be utilized to rescue other dystrophic skeletal muscle.

In the current study, we used a *desmin* knockout zebrafish line to identify a subset of genes specifically upregulated in EOMs, via transcriptome analysis. We further investigated one of these genes, *fhl2b*, and demonstrate that *fhl2b* has an important role in the inherent resistance of EOMs towards muscular dystrophies. Additionally, we show that *fhl2b* significantly improves survival, muscle integrity, and function of *sapje* zebrafish making it a novel target in the treatment of muscular dystrophies.

## Results

### A zebrafish model of desminopathy displays skeletal muscle defects

To study the EOMs in a non-lethal muscular dystrophy context, we generated *desma*[+/-]*;desmb*[+/-] zebrafish with premature stop codons in exon 1 of both genes (Fig. S1a, b). These mutations lead to truncated desmin lacking α-helix-rod domains, essential for coil formation and tertiary structure formation. We confirmed that the *desma*[-/-]*;desmb*[-/-] double mutants, unlike *desma*[+/-]*;desmb*[+/-] controls, lacked desmin immunolabeling at 3 dpf (Fig. S1c). Next, we performed functional experiments to investigate the impact of lack of desmin in zebrafish. Larvae were reared in 1% methyl cellulose, to generate swimming resistance, between days 4 and 5. This triggered significant myofiber detachment and breaks, both in *desma*[-/-]*;desmb*[-/-] mutants and *desma*[-/-] single mutants, at the 10–12th somite level (Fig. 1a, b), whereas *desmb*[-/-] and *desma*[+/-]*;desmb*[+/-] displayed no myofiber

damage (Fig. 1b), indicating that *desma* is the main contributor to muscle tensile strength. Previous studies have shown both *desma* and *desmb* expression in developing zebrafish somite musculature[31]. Therefore, to investigate the complete knockout of *desmin* genes and avoid potential confounding factors, we decided to continue our studies using only *desma*[-/-]*;desmb*[-/-] zebrafish. Force/tension relationship measurements revealed that 5-6 dpf *desma*[-/-]*;desmb*[-/-] zebrafish larvae display a significant decrease in trunk myofiber maximal force generation when compared to *desma*[+/-]*;desmb*[+/-] (Fig. 1c, d). Furthermore, *desma*[-/-]*;desmb*[-/-] mutants showed a significant decrease in spontaneous movement in comparison to *desma*[+/-]*;desmb*[+/-] larvae (Fig. 1e, f). This was not due to myofiber loss, as both genotypes had equal numbers of *Tg(mylz2:EGFP)* fast and *Tg(smyhc1:tdTomato)* slow twitch positive myofibers in cross sections of the trunk (Fig. S1d, e). Altogether, these results show that desmin contributes to myofiber integrity, and is needed to maintain proper function of muscle tissue in the embryonic zebrafish trunk.

### Extraocular muscle integrity is preserved in the zebrafish desminopathy model

Patients with desminopathy are often asymptomatic until mid-30s[29]. Therefore, to characterize our desminopathy model at later stages, we reared *desma*[-/-]*;desmb*[-/-] mutants and controls under equal conditions until 20–24 months of age, approximately two thirds of domesticated zebrafish life span, at which point they were histologically analyzed. *desma*[-/-]*;desmb*[-/-] mutants were fertile and survived until adulthood when reared separately from siblings (Fig. S1f). A consistent feature of muscular dystrophies is a glycolytic shift in myofiber identity from fast to slow[32,33]. To assess whether the lack of desmin had an impact on myofiber identity, trunk and EOMs of 24 months old zebrafish were immunolabeled using antibodies against fast myosin light chain (F310) and against slow myosin heavy chain 1, 2 and 3 (S58)[34]. The zebrafish trunk consists of myofibers subdivided into compartments separated by thin layers of connective tissue, myosepta, easily identified on cross-sections (Fig. 1, dashed line in g, i, k, m). Slow myofibers are generally found in the most laterally positioned compartments, separated from fast myofibers by a hyperplastic growth zone positioned near the myosepta[35]. Adult zebrafish EOMs also display a distinct location of slow and fast myofibers, although not separated by a myoseptum[20] (Fig. 1o, q). Cross-sections of trunk muscle of *desma*[-/-]*;desmb*[-/-] mutants revealed a decrease of fast F310 positive myofibers inside the slow domain (Fig. 1g, h, open arrowheads), a decrease in the slow myofiber diameter together with a significant increase in myofiber number in the slow domain, compared to wild type (WT) controls (*desma*[+/+]*;desmb*[+/+]) (Fig. 1l, j). Additionally, the number of slow myofibers inside the fast-medial compartments was significantly increased (Fig. 1l, j, arrowheads). Cross-sections of EOMs showed that they remained unaffected in terms of proportion of fast and slow myofibers (Fig. 1o–r). In summary, adult *desma*[-/-]*;desmb*[-/-] mutant trunk muscles show a glycolytic-fast to oxidative-slow metabolic shift, consistent with a muscular dystrophy phenotype whereas the EOMs remain unaffected in this regard.

To further define the *desma*[-/-]*;desmb*[-/-] muscle phenotype, we analyzed zebrafish trunk muscles and EOMs for signs of regeneration, proliferation and cell death. *desma*[-/-]*;desmb*[-/-] mutant trunk muscle showed a significantly increased proportion of both Pax7 positive nuclei (Fig. 1k, l) and PCNA positive nuclei (Fig. 1m, n) in the slow myofiber domains compared to controls. No change in TUNEL positive nuclei in the slow domains was observed, however, the fast myofiber domains contained clusters of myofibers where most nuclei were TUNEL positive (Fig. S1g, arrows), along with a significant increase in centrally positioned nuclei, a hallmark of muscular dystrophy (Fig. S1g, h, asterisk). In contrast, controls only appeared to have sporadic signs of DNA fragmentation dispersed among nuclei in the entire myofiber population (Fig. S1g). Interestingly, in EOM cross

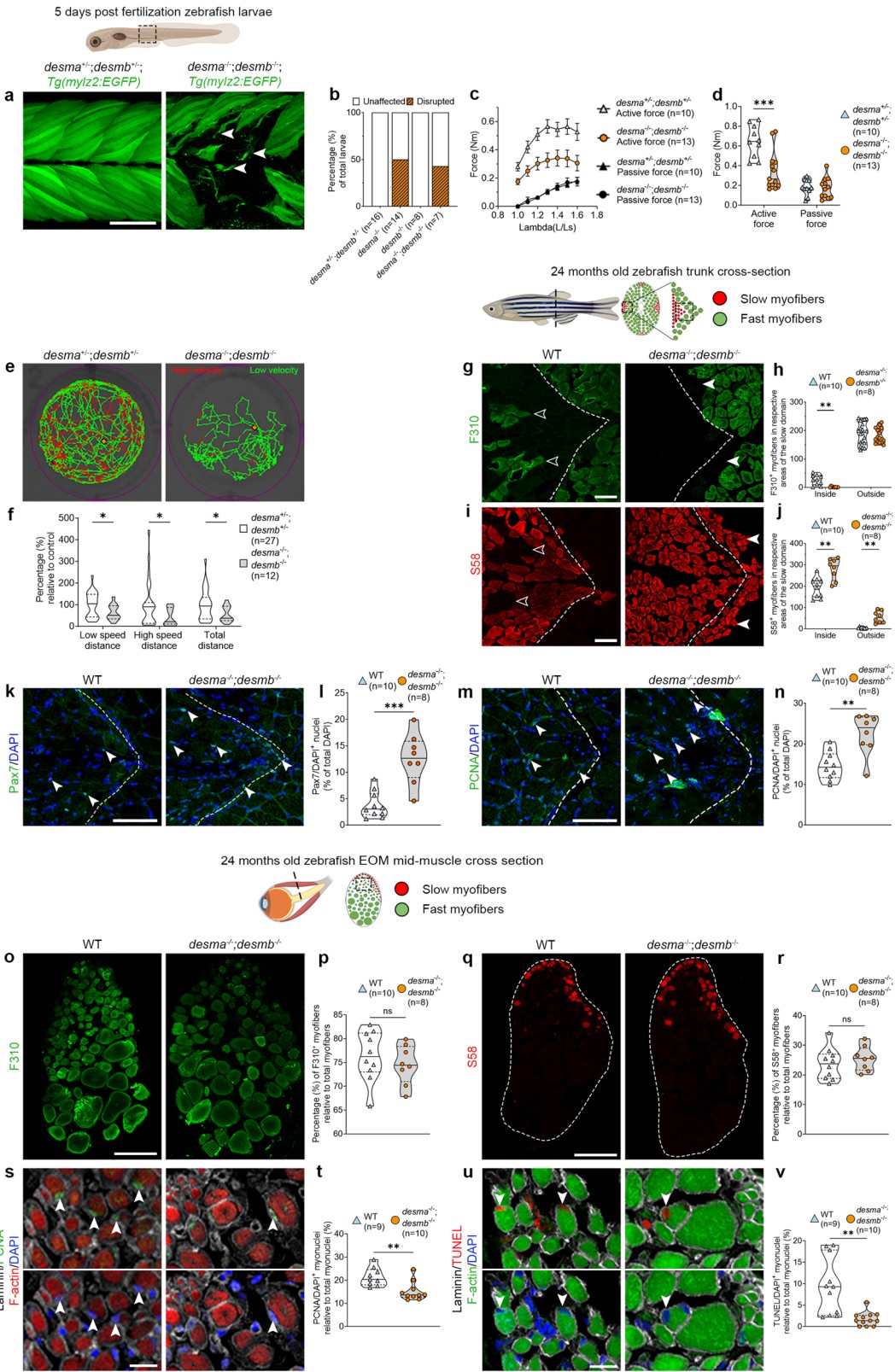

sections of $desma^{-/-};desmb^{-/-}$ mutants, we instead noted a significant decrease of PCNA positive myonuclei (Fig. 1s, t) and TUNEL positive myonuclei (Fig. 1u, v) compared to controls, suggesting a decreased cellular turn over in response to the lack of desmin in the EOMs. Overall, EOM myofiber composition remains unchanged despite the lack of desmin, whereas trunk muscle is significantly affected and shows clear signs of muscular dystrophy.

**Upregulation of *fhl2b* is an EOM response to muscular dystrophy**

To investigate the adaptations observed in desmin deficient EOMs, we performed RNA-sequencing of EOMs and trunk muscles at 5 and 20 months of age, representing pre- and symptomatic stages, respectively. For each stage, we obtained transcriptome profiles from both trunk muscles and EOMs from $desma^{-/-};desmb^{-/-}$ mutant and WT controls (Fig. S2a, schematic illustration). To identify genes involved

**Fig. 1 | Lack of desmin causes myofiber impairment and a metabolic shift in trunk myofibers. a** Tg(mylz2:EGFP);desma[−/−];desmb[−/−] and desma[+/−];desmb[+/−] larvae exposed to resistance swimming during 12 h from 4 to 5 dpf. Arrowheads: myofiber breaks. **b** Proportions of injuries caused by resistance swimming. **c** Force generation of desma[+/−];desmb[+/−] and desma[−/−];desmb[−/−] trunk myofibers. **d** Force generated at optimal stretch in desma[+/−];desmb[+/−] and desma[−/−];desmb[−/−] (p = 0.002). **e** Representative swimming tracks of desma[−/−];desmb[−/−] and desma[+/−];desmb[+/−] controls. **f** Relative swimming distance at low speed (p = 0.024), high speed (p = 0.019) and total distance (p = 0.016). **g** F310 trunk immunolabeling of desma[−/−];desmb[−/−] and WT. Open arrowheads: F310[+] myofibers inside the slow domain, arrowheads: lack of F310 labeling in the fast domain. **h** Comparison of number of fast myofibers inside (p = 0.0018) and outside the slow domain. **i** S58 trunk immunolabeling in desma[−/−];desmb[−/−] and WT. Open arrowheads: lack of slow myofibers in the slow domain, arrowheads: S58[+] myofibers in the fast domain. **j** Quantification of number of slow myofibers inside (p = 0.002) and outside (p = 0.0012) of the slow domain. **k** WT and desma[−/−];desmb[−/−] trunk immunolabeled for DAPI/Pax7. Arrowheads: Pax7[+] nuclei. **l** Quantification of Pax7[+]/DAPI[+] cells (p = 0.0008). **m** Cross-sections of WT and desma[−/−];desmb[−/−] trunk immunolabeled for DAPI/PCNA. Arrowheads: PCNA[+] nuclei. **n** Quantification of PCNA[+]/DAPI[+] cells (p = 0.0028). **o** Cross-sections of WT and desma[−/−];desmb[−/−] EOMs immunolabeled for F310. **p** Quantification of F310[+] myofibers. **q** Cross-sections of WT and desma[−/−];desmb[−/−] EOMs immunolabeled for S58. **r** Quantification of S58[+] myofibers. **s** DAPI/PCNA/laminin/phalloidin immunolabeling of desma[−/−];desmb[−/−] and WT EOMs. Arrowheads: DAPI[+]/PCNA[+] myonuclei. **t** Quantification of DAPI/PCNA[+] myonuclei (p = 0.0039). **u** DAPI/TUNEL/laminin/phalloidin immunolabeling of desma[−/−];desmb[−/−] and WT EOMs. Arrowheads: DAPI[+]/TUNEL[+] myonuclei. **v** Quantification of DAPI[+]/TUNEL[+] myonuclei (p = 0.0032). Statistical analysis: Two-sided t-tests with Welch correction. Data in violin plots is presented as median (line) and quartiles (dashed lines). Data in (**c**) is presented as mean ± SEM. Dashed lines in cross-sections: myosepta separating fast and slow domains. Age of fish, tissue analyzed, viewed area and level of cross-section is illustrated above panels. Scale bars in **a, o, q**: 100 μm, **g, i, k, m**: 50 μm, **s, u**: 25 μm. Schematic images were adapted from https://www.biorender.com.

specifically in the EOMs resistance towards muscular dystrophy, we performed differential expression analysis in desma[−/−];desmb[−/−] compared to WT controls in EOMs (Fig. 2a, comparison I-II) and trunk (Fig. 2a, comparison III-IV) at both time points (Fig. S2a–e, Supplementary Data 1), additionally, we also compared EOMs to trunk within the same genetic background (Fig. 2, comparisons V-VIII, Fig. S2f–i, Supplementary Data 1). Subsequently, we performed Gene Ontology (GO) analysis and retrieved differentially expressed genes (DEGs) contained within the cellular compartment GO terms related to myofiber function/structure (Fig. S3a–f, Supplementary Data 2), which we further cross-compared (Fig. 2a, comparisons I-VIII). Interestingly, when analyzing 20-month-old EOMs, we found upregulation of several DEGs related to cytoskeletal rearrangement (Fig. 2a, comparison III). One of these genes, fhl2b, was consistently identified across comparisons III-VIII highlighting it as a potential candidate gene (Fig. 2a). Notably, four members of the fhl family, including fhl2b, were identified in all EOM vs trunk comparisons (Fig. 2a). We did not find fhl2b to be differentially expressed in 5 months old desma[−/−];desmb[−/−] EOMs which can likely be attributed to the early stage of disease progression. Next, we analyzed fhl2b gene expression across the different comparisons (Fig. 2a, comparisons I-VIII) and found that fhl2b consistently was more highly expressed in EOMs compared to trunk muscle (Fig. 2b). Previous studies have shown that Fhl2 is mainly expressed in cardiac muscle[36], localized to Z-discs, but has to our knowledge not been studied in the EOMs prior to this study. EOM longitudinal sections immunolabeled for Fhl2 showed localization to the Z-disc (Fig. S3g). The number of Fhl2-immunolabelled myofibers in EOMs of desma[−/−];desmb[−/−] mutants was significantly increased relative to controls at both 5 and 20 months of age and increased significantly with age (Fig. 2c–e) confirming our transcriptomic analysis. Next, we asked whether an increase in Fhl2 positive myofibers of EOMs could be found across muscular dystrophies and therefore immunolabeled plecb[−/−] (plectin) and obscnb[−/−] (obscurin) mutant zebrafish EOMs (Fig. 2f), both resulting in muscular dystrophy when mutated in other models[37,38]. Interestingly, plecb[−/−] and obscnb[−/−] mutant zebrafish displayed increased numbers of Fhl2 positive myofibers compared to controls, similar to desma[−/−];desmb[−/−] EOMs (Fig. 2c, d, f). This shows that Fhl2 is more widespread in several different cytoskeletal gene mutations and models for muscular dystrophy. Importantly, healthy adult human and mouse EOMs were also found to contain FHL2-positive myofibers (Fig. 2g, i), confirmed by western blots (Fig. 2h, j), indicating a putatively conserved role for Fhl2 in EOMs across species. In summary, Fhl2 is present in EOMs across different muscular dystrophies and species and is increased with the progression of the disease in the EOMs of desma[−/−];desmb[−/−] mutant zebrafish.

## Knockout of *fhl2* leads to EOM myofiber hypertrophy

Previous studies have found Fhl2 to be mainly expressed in cardiac muscle, however, Fhl2 knockout mice were shown to be viable and no phenotype was observed unless challenged with isoproterenol, triggering adrenergic stimuli which resulted in cardiac hypertrophy[39]. We hypothesized that knockout of fhl2 in the background of desma[−/−];desmb[−/−] mutations could cause similar phenotypes in EOMs of adult zebrafish. We generated knockout lines of both zebrafish fhl2 genes, fhl2a, and fhl2b, to avoid possible confounding effects of redundancy (Fig. S4a, b). In situ hybridization of fhl2a showed low EOM expression whereas fhl2b was found to be distinctly expressed in the EOMs at 5 dpf (Fig. S4c, d, arrowheads) suggesting a larger role for fhl2b compared to fhl2a in the EOMs. Immunolabeling using Fhl2 antibodies confirmed lack of Fhl2 in desma[−/−];desmb[−/−];fhl2a[−/−];fhl2b[−/−] compared to desma[−/−];desmb[−/−];fhl2a[+/−];fhl2b[+/−] sibling controls (Fig. S4e). To further elucidate the effect of lack of fhl2 genes on other fhl family members, we analyzed the gene expression of fhl1a, fhl1b, fhl3a and fhl3b in the different mutant larvae. There were no significant differences between WT and desma[−/−];desmb[−/−] double mutants for these genes, however, we found that knockout of fhl2a, fhl2b or both genes rendered a general downregulation of fhl1a and fhl1b (Fig. S4f, g). fhl3a expression remained unchanged across all comparisons, however, a similar, less pronounced, downregulation trend was observed for fhl3b (Fig. S4h, i). These results are in line with previous findings[40], and indicate that knockout of fhl2 genes is not compensated by other fhl family members. Instead, our results imply that the expression of fhl1 and fhl3 may be partly dependent on Fhl2. In addition, previous studies have linked Fhl2 to WNT signaling via β-catenin[41–44] and we therefore investigated if knockout of fhl2 genes would affect this signaling pathway. Similar to the previous analysis, we found no significant differences between WT and desma[−/−];desmb[−/−] double mutants. Our results showed that knockout of either fhl2a, fhl2b or both fhl2a and fhl2b significantly altered the levels of wnt5a, wnt5b, wnt11, and ctnnb2 (Fig. S4j–n). These data further support a role for Fhl2 in the WNT/ β-catenin signaling pathway.

Quantification of EOM myofiber size at 12 months of age in single fhl2a[−/−] and fhl2b[−/−] mutants and fhl2a[−/−];fhl2b[−/−] double mutants, all in a desma[−/−];desmb[−/−] background, alongside the corresponding controls revealed that the myofiber areas were significantly increased in desma[−/−];desmb[−/−];fhl2b[−/−] and desma[−/−];desmb[−/−];fhl2a[−/−];fhl2b[−/−] mutants compared to WT controls (Fig. 3a, b). This indicates that fhl2b has a role in hypertrophic protection, in line with results observed for Fhl2 in mouse cardiac muscle[39]. Given the reduction on cell death and proliferation observed in desma[−/−];desmb[−/−] mutant EOMs (Fig. 1s–v), we wondered whether the knockout of fhl2a and fhl2b would revert these effects. For this purpose, we addressed cell death and proliferation on EOM cross sections. TUNEL (Fig. 3c–e) and PCNA (Fig. 3f–h) labeling of myonuclei showed a redundant relationship between fhl2a and fhl2b. Both desma[−/−];desmb[−/−];fhl2a[−/−] and desma[−/−];desmb[−/−];fhl2b[−/−] triple mutants displayed no change in cell death whereas a moderate increase in proliferation was found. However, in desma[−/−];desmb[−/−];fhl2a[−/−];fhl2b[−/−]

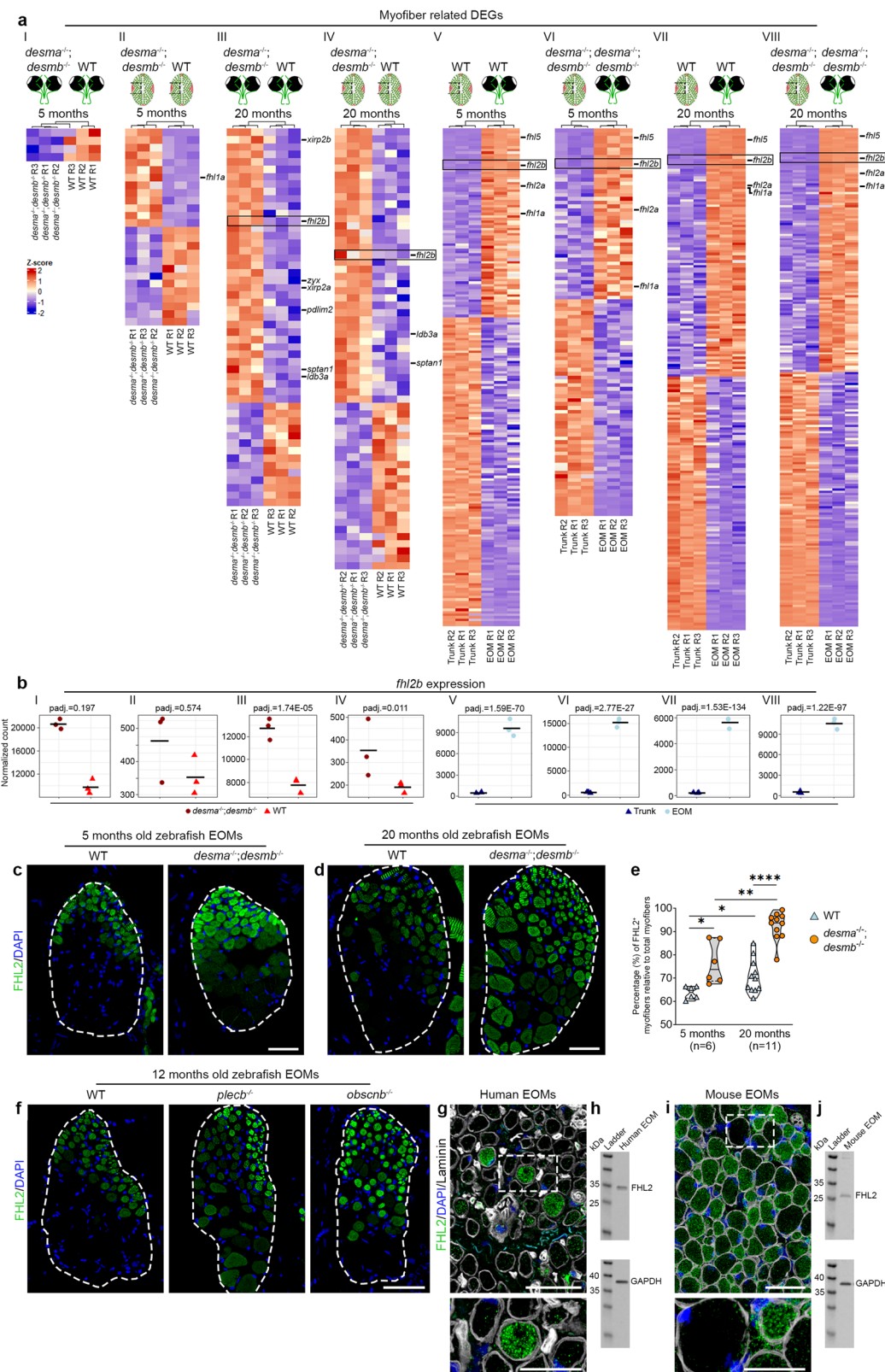

**a** Myofiber related DEGs

**b** *fhl2b* expression

**c** 5 months old zebrafish EOMs

**d** 20 months old zebrafish EOMs

**f** 12 months old zebrafish EOMs

**g** Human EOMs

**i** Mouse EOMs

quadruple mutant zebrafish EOMs, a highly significant increase in both cell death and proliferation was observed (Fig. 3e, h). Given that hypertrophic myofibers were observed in *desma⁻/⁻;desmb⁻/⁻;fhl2b⁻/⁻* and *desma⁻/⁻;desmb⁻/⁻;fhl2a⁻/⁻;fhl2b⁻/⁻* EOMs at 12 months, we further analyzed 12-month-old cardiac and skeletal muscle tissues for hypertrophy in our mutants, however, no significant changes were observed at this stage (Fig. S5a, h). These results indicate that both *fhl2a* and *fhl2b*

maintain myonuclei integrity in the EOMs under muscle dystrophy conditions, however, *fhl2b* likely has an additional role in hypertrophic protection given the increase in myofiber area observed in the examined mutants lacking *fhl2b* (Fig. 3a, b). Collectively, we show that Fhl2 is needed to maintain myofiber homeostasis in the EOMs in *desmin* knockout background and can therefore be considered as an EOM protective gene.

**Fig. 2 | *fhl2b* is upregulated in the EOM in response to desmin-related muscular dystrophy. a** Expression of myofiber-related DEGs for the following comparisons: 5 months *desma^-/-^:desmb^-/-^* vs WT EOMs (I), 5 months *desma^-/-^:desmb^-/-^* vs WT trunk (II), 20 months *desma^-/-^:desmb^-/-^* vs WT EOMs (III), 20 months *desma^-/-^:desmb^-/-^* vs WT trunk (IV), 5 months WT EOMs vs WT trunk (V), 5 months *desma^-/-^:desmb^-/-^* EOMs vs *desma^-/-^:desmb^-/-^* trunk (VI), 20 months WT EOMs vs WT trunk (VII) and 20 months *desma^-/-^:desmb^-/-^* EOMs vs *desma^-/-^:desmb^-/-^* trunk (VIII). **b** *fhl2b* expression in the abovementioned comparisons I-VIII. FHL2 antibody labeling of WT and *desma^-/-^;desmb^-/-^* cross-sectioned EOMs at **c** 5 and **d** 20 months. **e** FHL2 positive myofibers quantification of *desma^-/-^;desmb^-/-^* versus WT control EOMs in 5 ($p = 0.017$) and 20-monthold zebrafish ($p = 3.7e^{-7}$), respectively and 5-months versus 20-months-old WT ($p = 0.014$) and *desma^-/-^:desmb^-/-^* ($p = 0.005$), respectively. Data is presented as median (line) and quartiles (dashed lines). **f** WT, *plecb^-/-^* and *obscnb^-/-^* 12-month-old EOM cross sections immunolabeled with FHL2 antibodies. **g** Human EOM cross section immunolabeled with DAPI/FHL2/laminin antibodies, dashed square indicates area enlarged below. **h** Western blot on human EOMs showing FHL2. **i** Mouse EOM cross section immunolabeled using DAPI/FHL2/laminin antibodies, dashed boxes indicate area enlarged below. **j** Western blot of mouse EOMs showing FHL2. Statistical analysis in b: Two-sided Wald test with B/H-correction, e: Two-sided t-tests with Welch correction. Scale bars in c, d, f, g, i: 50 μm, g, i bottom panel: 25 μm. White dashed lines outline the entire cross-section of the EOMs. Schematic images were adapted from https://www.biorender.com.

## Muscle specific overexpression of *fhl2b* significantly improves survival and myofiber integrity of the Duchenne muscular dystrophy zebrafish model

Since *fhl2b* was the only Fhl2 paralog differentially expressed in our transcriptomic data of EOMs lacking *desmin*, we chose to investigate whether *fhl2b* also could protect muscles other than EOMs in muscular dystrophy. To test this in a more severe context, we utilized the lethal zebrafish *sapje* (*dmd^ta222a^*) line[9], a model of Duchenne muscular dystrophy, hereby referred to as *dmd^-/-^*. We overexpressed *fhl2b* under the muscle-specific 503unc promoter[45] coupled to EGFP via a T2A linker and analyzed trunk muscle in the *dmd^-/-^* background (Fig. 4a). We initially examined mosaic overexpression of *fhl2b*, which showed an overlap between EGFP positive myofibers and Fhl2 antibody positive myofibers (Fig. S6a). *dmd^-/-^* larvae with a high number of EGFP positive myofibers generally showed less myofiber disruption than seen in *dmd^-/-^* larvae with low number EGFP positive fibers (Fig. S6b). We therefore reared mosaic larvae to adulthood and generated stable overexpression transgenic lines from three different founders with different EGFP intensity levels, one low and two high (Fig. S6c). To functionally test the effects of *fhl2b* overexpression on *dmd^-/-^* larvae, we analyzed survival among low and high level *fhl2b* expressing lines and found a significant correlation between *fhl2b* expression (Fig. S4c) and survival of *dmd^-/-^;Tg(503unc:fhl2b-T2A-EGFP)* larvae (Fig. 4b). The median survival time of *dmd^-/-^* larvae was found to be 18 dpf, whereas *dmd^-/-^;Tg(503unc:fhl2b-T2A-EGFP)* and sibling control median survival time was longer than 30 days (Fig. 4b). Notably, even low levels of *fhl2b* overexpression increased survival beyond 30 days in *dmd^-/-^;Tg(503unc:fhl2b-T2A-EGFP)*. Additionally, spontaneous swimming distance was also significantly increased in *dmd^-/-^;Tg(503unc:fhl2b-T2A-EGFP)* larvae compared to *dmd^-/-^* larvae at 5 dpf (Fig. 4c, d). Collectively, these results show that muscle specific *fhl2b* overexpression improves motor function and significantly prolongs the lifespan in this model of Duchenne muscular dystrophy.

To examine if *fhl2b* overexpression improved myofiber integrity, we labeled 5 dpf *dmd^-/-^;Tg(503unc:fhl2b-T2A-EGFP)*, *dmd^-/-^* and sibling control (*dmd^+/+^, dmd^+/-^*) larvae using phalloidin and DAPI. Additionally, we generated a *Tg(503unc:EGFP)* line which was crossed into the *dmd^+/-^* line, used as a negative control. *dmd^-/-^;Tg(503unc:fhl2b-T2A-EGFP)* larvae showed detached myofibers similar to *dmd^-/-^* and *dmd^-/-^;Tg(503unc:EGFP)* larvae (Fig. 4e, f). However, *dmd^-/-^* and *dmd^-/-^;Tg(503unc:EGFP)* detached myofibers left gaps in the muscle devoid of F-actin (Fig. 4f, closed arrowheads). Interestingly, detached myofibers in *dmd^-/-^;Tg(503unc:fhl2b-T2A-EGFP)* myotomes seldom had these gaps and thin F-actin positive myofibers were present (Fig. 4f, open arrowheads) which exhibited a high EGFP intensity, suggesting that they are newly formed as the 503unc promoter is more active in early stages of myogenesis[46] (Fig. 4g). We quantified myofiber detachment in *dmd^-/-^* and *dmd^-/-^;Tg(503unc:fhl2b-T2A-EGFP)* larvae using phalloidin. We found that *dmd^-/-^;Tg(503unc:fhl2b-T2A-EGFP)* consistently showed reduced number of myofiber detachments per somite compared to *dmd^-/-^* controls (Fig. 4h, i). Overall, these data indicate that *dmd^-/-^;Tg(503unc:fhl2b-T2A-EGFP)* myofibers have a lower tendency to detach compared to *dmd^-/-^* myofibers.

## Muscle specific *fhl2b* overexpression improves motor axon integrity and neuromuscular junctions in *dmd^-/-^* larvae

To further understand how overexpression of *fhl2b* improves the *dmd^-/-^* phenotype, we analyzed the transcriptomes of trunk muscle tissue from sibling controls (*dmd^+/+^, dmd^+/-^*), sibling controls with *fhl2b* overexpression (*Tg(503unc:fhl2b-T2A-EGFP)*), *dmd^-/-^* and *dmd^-/-^;Tg(503unc:fhl2b-T2A-EGFP)* larvae using RNA-sequencing at 5 dpf (Fig. S6d–j, Supplementary Data 3). We then intersected the DEGs from the pairwise comparisons *dmd^-/-^* vs sibling controls and *dmd^-/-^;Tg(503unc:fhl2b-T2A-EGFP)* vs sibling controls and identified 1054 DEGs unique to *dmd^-/-^* larvae (Fig. 5a, gene set C). These 1054 DEGs correspond to *dmd* disease-related genes, whose expression is partially rescued by *fhl2b* overexpression (Fig. 5b). Gene ontology analysis of all intersections (Fig. 5c, Fig. S7a, b) revealed a unique enrichment for axon and neuron guidance-related terms in *dmd^-/-^* larvae (Fig. 5c), which notably included several *semaphorin* genes, crucial for axon guidance and motor neuron survival (Fig. 5d). In addition, *mir-206*, a regulatory microRNA suggested to mediate communication between myofibers and neurons, and histone deacetylase 4 (*hdac4*) are both upregulated in Duchenne muscular dystrophy patients and suggested as biomarkers[47,48]. We determined the expression of these genes in our system and observed that they were significantly upregulated in *dmd^-/-^* larvae (Fig. 5e, f). However, in *dmd^-/-^;Tg(503unc:fhl2b-T2A-EGFP)* larvae, the expression levels of both genes were significantly closer to that of sibling controls, which comparatively might be indicative of healthier tissue (Fig. 5e, f). To test whether axon and neuromuscular junction (NMJ) integrity were improved in *dmd^-/-^;Tg(503unc:fhl2b-T2A-EGFP)* larvae, we immunolabeled 5 dpf larvae using acetylated tubulin antibodies and α-bungarotoxin, labeling axons and post-synaptic NMJs, respectively (Fig. 5g–i, Fig. S8). *dmd^-/-^* larvae displayed thin axons with reduced branching compared to sibling controls (Fig. 5g, i, arrowhead), additionally, NMJs were disorganized and fragmented (Fig. 5h, i arrow) and an almost complete loss of contact was registered between axons and NMJs. In contrast, *dmd^-/-^;Tg(503unc:fhl2b-T2A-EGFP)* larvae displayed near normal axonal branching (Fig. 5g) and NMJ organization (Fig. 5i, open arrowhead). Because the *dmd^-/-^* NMJs were extremely fragmented, we defined healthy NMJs as α-bungarotoxin positive puncta larger than the smallest WT puncta. The healthy NMJs were significantly reduced in *dmd^-/-^*, but not in *dmd^-/-^;Tg(503unc:fhl2b-T2A-EGFP)* larvae (Fig. 5k). Additionally, double positive Synaptic vesicle 2 (SV2)/α-bungarotoxin NMJs were significantly fewer in *dmd^-/-^* as compared to *dmd^-/-^;Tg(503unc:fhl2b-T2A-EGFP)* and sibling control larvae (Fig. 5j arrowhead, Fig. 5l), indicating decreased innervation of myofibers. To quantify axon integrity, we measured the length (Fig. 5m, o) and volume (Fig. 5n, p) of axons using 3D-renderings of acetylated tubulin immunolabeled larvae and found significant improvements in *dmd^-/-^;Tg(503unc:fhl2b-T2A-EGFP)* compared to *dmd^-/-^* larvae. Together, our data indicate that axon retraction or degradation is a commonly occurring phenomenon in *dmd^-/-^* larvae. Furthermore, our data suggest that in myofibers rescued by *fhl2b* expression present a significant improvement in axons, NMJs, innervation and functionality as evidenced by swimming capacity (Fig. 4d). However, we recognize that this may be the consequence of improved muscle tissue integrity.

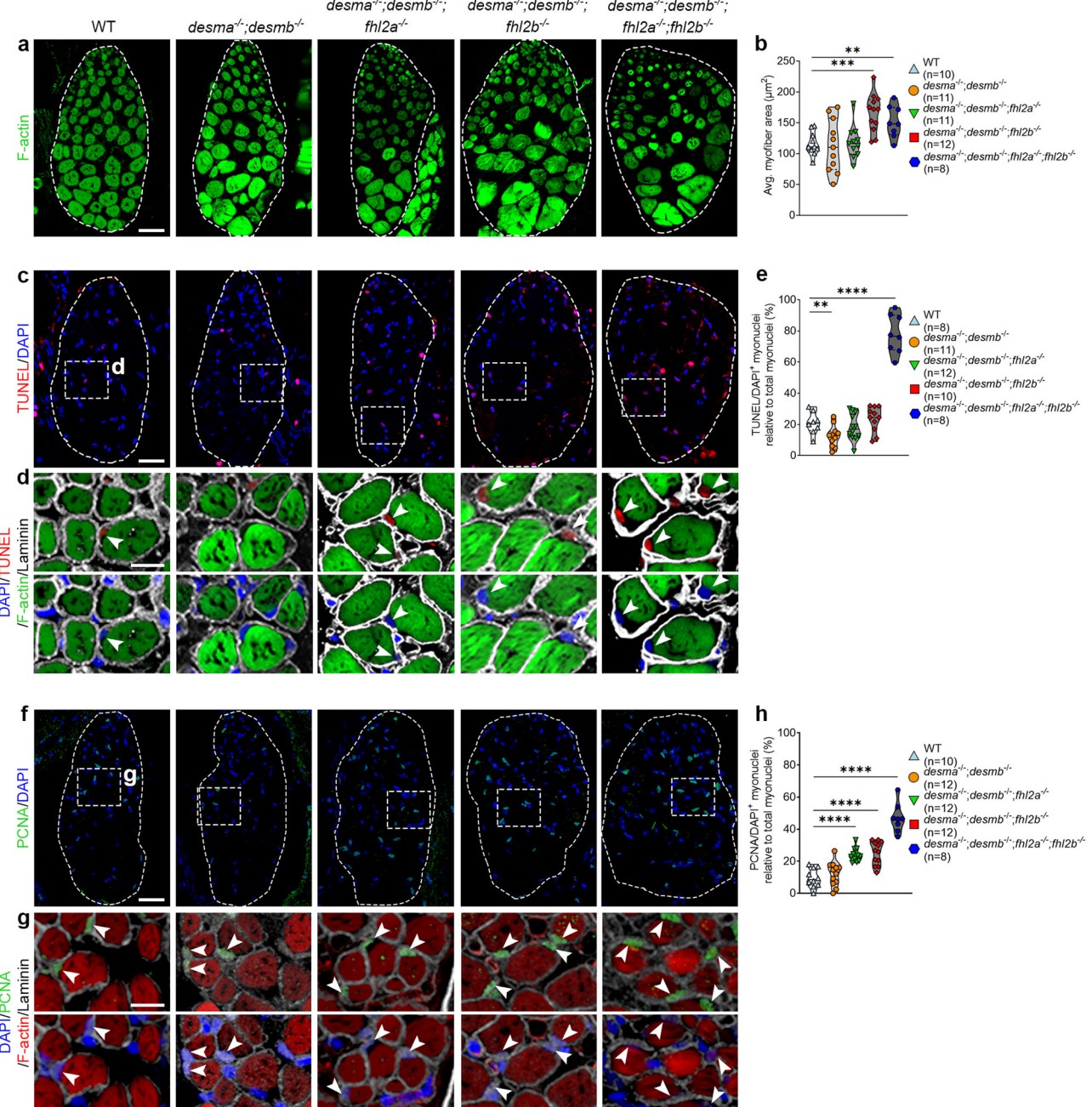

**Fig. 3 | Lack of Fhl2 causes EOM myofiber hypertrophy.** Cross-sections of 12 months old WT, *desma⁻/⁻;desmb⁻/⁻*, *desma⁻/⁻;desmb⁻/⁻;fhl2a⁻/⁻*, *desma⁻/⁻;desmb⁻/⁻;fhl2b⁻/⁻* and *desma⁻/⁻;desmb⁻/⁻;fhl2a⁻/⁻; fhl2b⁻/⁻* zebrafish EOMs. **a** F-actin labeling (phalloidin) and **b** average F-actin positive myofiber area quantification in WT, *desma⁻/⁻;desmb⁻/⁻*, *desma⁻/⁻;desmb⁻/⁻;fhl2a⁻/⁻*, *desma⁻/⁻;desmb⁻/⁻;fhl2b⁻/⁻* (*p* = 0.0002) and *desma⁻/⁻;desmb⁻/⁻;fhl2a⁻/⁻;fhl2b⁻/⁻* (*p* = 0.007) mutant zebrafish. **c** DAPI and TUNEL labeling of myonuclei. Dashed boxes indicate magnified areas in **d**). **d** TUNEL (top) and DAPI (bottom) labeling of myonuclei inside the myofiber laminin sheet. Phalloidin labels F-actin in the myofibers. White arrowheads indicate TUNEL positive myonuclei. **e** Quantification of DAPI⁺/TUNEL⁺ myonuclei in WT, *desma⁻/⁻;desmb⁻/⁻* (*p* = 0.021), *desma⁻/⁻;desmb⁻/⁻;fhl2a⁻/⁻*, *desma⁻/⁻;desmb⁻/⁻;* *fhl2b⁻/⁻* and *desma⁻/⁻;desmb⁻/⁻;fhl2a⁻/⁻;fhl2b⁻/⁻* (*p* = 2.2e⁻⁷) mutant zebrafish. **f** DAPI/PCNA labeling of myonuclei. Dashed boxes indicate magnified areas in g). **g** DAPI/PCNA labeling of myonuclei inside the myofiber laminin sheet. Phalloidin labels F-actin. White arrowheads indicate double positive myonuclei. **h** Quantification of PCNA⁺ myonuclei in WT, *desma⁻/⁻;desmb⁻/⁻*, *desma⁻/⁻;desmb⁻/⁻;fhl2a⁻/⁻* (*p* = 8.8e⁻⁵), *desma⁻/⁻;desmb⁻/⁻;fhl2b⁻/⁻* (*p* = 2.3e⁻⁵), and *desma⁻/⁻;desmb⁻/⁻;fhl2a⁻/⁻;fhl2b⁻/⁻* (*p* = 4.6e⁻⁷) mutant zebrafish. Dashed lines outline the entire cross-section of the EOMs. Statistical analysis: Two-sided t-tests with Welch correction. Data in all violin plots is presented as median (line) and quartiles (dashed line). Avg average. Scale bar in **a, c, f**: 25 μm, **d, g**: 10 μm.

## Muscle specific overexpression of *fhl2b* accelerates macrophage activity and muscle regeneration

To further define mechanisms underlying survival of *dmd⁻/⁻; Tg(SO3unc:fhl2b-T2A-EGFP)* larvae, we intersected DEGs from *dmd⁻/⁻; Tg(SO3unc:fhl2b-T2A-EGFP) vs dmd⁻/⁻* and *Tg(SO3unc:fhl2b-T2A-EGFP) vs* sibling controls (*dmd⁺/⁺, dmd⁺/⁻*) (Fig. 6a, Fig. S7c). We identified 78 DEGs from *dmd⁻/⁻; Tg(SO3unc:fhl2b-T2A-EGFP) vs dmd⁻/⁻* and *Tg(SO3unc:fhl2b-T2A-EGFP) vs*

genes differentially expressed in response to *fhl2b* overexpression in the *dmd⁻/⁻* disease background (Fig. 6a, gene set F). Among these 78 DEGs, we found several genes involved in muscle regeneration (Fig. 6b, *flnca, rbfoxl1, ptgs2b, kdm6ba, mmp14b*), suggesting that this is a possible mechanism for improved muscle integrity in *dmd⁻/⁻; Tg(SO3unc:fhl2b-T2A-EGFP)* larvae. In addition, we analyzed 5 dpf

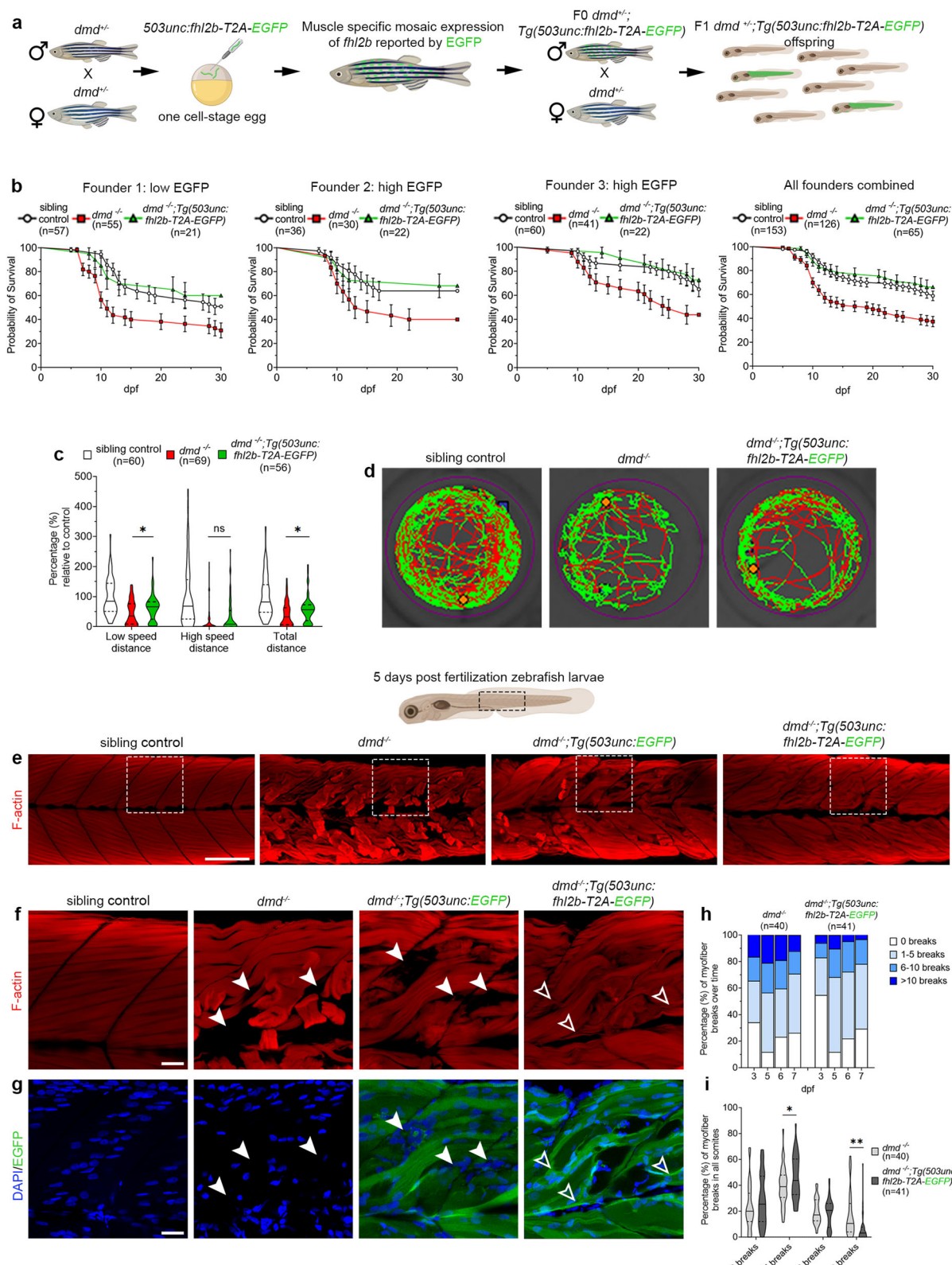

$dmd^{-/-};Tg(503unc:fhl2b-T2A-EGFP)$, $dmd^{-/-}$, $dmd^{-/-};Tg(503unc:EGFP)$ and sibling controls for muscle regeneration, proliferation and cell death (Fig. S9a–g). Interestingly, we noted a significant decrease in Pax7 (Fig. S9a, b), BrdU (24 h pulse, Fig. S9c–e) and TUNEL (Fig. S9f–g) positive nuclei in $dmd^{-/-};Tg(503unc:fhl2b-T2A-EGFP)$ compared to $dmd^{-/-}$ larvae, indicating overall healthier myofibers. These data confirm that $dmd^{-/-};Tg(503unc:fhl2b-T2A-EGFP)$ myofibers are partially

rescued by $fhl2b$ overexpression and support our findings that $dmd^{-/-};Tg(503unc:fhl2b-T2A-EGFP)$ myofibers are less prone to detachment. Interestingly, our study indicates that $fhl2b$ induces faster formation of new myofibers and we therefore hypothesized that the muscle regeneration would be quicker in $fhl2b$ overexpressing larvae. To address this, we performed timelapse studies on $dmd^{-/-};Tg(503unc:EGFP)$ and $dmd^{-/-};Tg(503unc:fhl2b-T2A-EGFP)$ larvae

**Fig. 4 | Muscle specific overexpression of *fhl2b* significantly prolongs life-span and improves motor function and muscle integrity in *dmd*⁻/⁻ zebrafish larvae.** **a** Experimental setup to generate *dmd*⁻/⁻ zebrafish overexpressing *fhl2b*. One cell-stage eggs from in-crossed *dmd*⁺/⁻ zebrafish were injected with a *503unc:fhl2b-T2A-EGFP* plasmid, raised and crossed into *dmd*⁺/⁻ to generate stable lines. **b** Survival tests over 30 days for three different *503unc:fhl2b-T2A-EGFP* lines. Kaplan-Meier log rank test was used to calculate significance between *dmd*⁻/⁻ and *dmd*⁻/⁻:*Tg(503unc:fhl2b-T2A-EGFP)* for each of the three founder lines (Founder 1: *p* = 0.0228, Founder 2: *p* = 0.0534, Founder 3: *p* = 0.0188 and all founder lines combined: *p* < 0.0001). **c** Spontaneous swimming tests showed significant increases in low speed (*p* = 0.0459) and total distance (*p* = 0.0385) in *dmd*⁻/⁻:*Tg(503unc:fhl2b-T2A-EGFP)* compared to *dmd*⁻/⁻ larvae. **d** Representative swimming tracks of sibling control, *dmd*⁻/⁻ and *dmd*⁻/⁻:*Tg(503unc: fhl2b-T2A-EGFP)* larvae. **e** F-actin (phalloidin) labeling of trunk muscle in sibling control, *dmd*⁻/⁻, *dmd*⁻/⁻:*Tg(503unc:EGFP)* and *dmd*⁻/⁻:

*Tg(503unc:fhl2b-T2A-EGFP)* larvae. **f** Magnification of dashed boxes in e) showing detached myofibers and empty areas in *dmd*⁻/⁻ and *dmd*⁻/⁻:*Tg(503unc:EGFP)* larvae (arrowheads) whereas *dmd*⁻/⁻:*Tg(503unc: fhl2b-T2A-EGFP)* larvae display **g** small diameter intensely EGFP positive myofibers (green) in corresponding areas (open arrowheads). DAPI in blue. **h** Quantification of detached F-actin⁺ myofibers in somite segments at 3, 5, 6 and 7 dpf. **i** Number of myofiber breaks per somite. In total, *dmd*⁻/⁻:*Tg(503unc: fhl2b-T2A-EGFP)* larvae show more small (1–5 breaks) myofiber detachments per somite (*p* = 0.0441) and less large (>10 breaks) myofiber detachments per somite (*p* = 0.001) as compared to *dmd*⁻/⁻ larvae. Statistical analysis in c, i: Two-sided t-tests with Welch correction. Data in violin plots (c, i) are presented as median (line) and quartiles (dashed line). Data in all survival graphs (**b**) are presented as mean ± SEM. Scale bar in **e**: 100 µm, **f**: 25 µm. Schematic images were adapted from https://www.biorender.com.

during 24 h to observe muscle regeneration in real time (Fig. 6c). We observed that detached myofiber debris was quickly removed and myofibers were apparently replaced in *dmd*⁻/⁻;*Tg(503unc:fhl2b-T2A-EGFP)* larvae whereas practically no change was observed in *dmd*⁻/⁻;*Tg(503unc:EGFP)* controls during the 24 h window (Fig. 6c). To quantify this, we measured muscle integrity using birefringence. We found a significant improvement in *dmd*⁻/⁻;*Tg(503unc:fhl2b-T2A-EGFP)* over time as compared to *dmd*⁻/⁻;*Tg(503unc:EGFP)* controls (Fig. 6d, e). Collectively, these results suggest that *fhl2b* overexpression accelerates muscle regeneration.

Given our results showing that detached myofibers quickly disappeared in *dmd*⁻/⁻;*Tg(503unc:fhl2b-T2A-EGFP)* larvae (Fig. 6c), we hypothesized that increased leukocyte activity could be involved in the muscle regeneration process. Leukocytes have previously been shown to be critical in zebrafish muscle regeneration[49]. Additionally, macrophage specific genes *mmp13a* and *mpeg1.2* as well as *il-8* were upregulated in *dmd*⁻/⁻;*Tg(503unc:fhl2b-T2A-EGFP)* and uninjured *Tg(503unc:fhl2b-T2A-EGFP)* as compared to sibling controls (Fig. 7a), supporting this hypothesis. To address this, we analyzed *fhl2b*´s effect on various aspects of muscle regeneration in a controlled setting using *Tg(503unc:fhl2b-T2A-EGFP)* larvae compared to sibling controls. We performed needle-stick injuries and analyzed the muscle regeneration process by immunolabeling for neutrophils (Mpx), macrophages (Mfap4) and satellite cells (Pax7) coupled with DAPI and phalloidin, over time (Fig. 7b). All three antibodies were found to label an equal number of cells in 3 dpf uninjured controls (Fig. S.9h-m), and the neutrophil population labeled by Mpx antibodies was unchanged across all stages tested, from uninjured to 24 h post injury (hpi) (Fig. 7c, d). Noticeably, we found a significant increase in macrophage count at the wound site in *Tg(503unc:fhl2b-T2A-EGFP)* larvae already at 6 hpi (Fig. 7e, f). This was followed by a significantly reduced number of macrophages at 24 hpi in *Tg(503unc:fhl2b-T2A-EGFP)* wounds compared to sibling control wounds (Fig. 7e, f). Pax7 positive satellite cells were present at wound sites faster in *Tg(503unc:fhl2b-T2A-EGFP)* compared to sibling controls and peaked at 48 hpi whereas sibling control satellite cell numbers peaked at 72 hpi (Fig. 7g, h). Additionally, F-actin intensity was found to be significantly higher at 72 hpi as compared to controls (Fig. 7I, j). Taken together, these results suggest that *Tg(503unc:fhl2b-T2A-EGFP)* larvae have faster myofiber regeneration via enhanced macrophage recruitment.

## Discussion

In the current study we show that induced expression of *fhl2b* can provide resistance to muscular dystrophy and that the EOMs offer a novel approach regarding protective cellular strategies. Here we propose a model (Fig. 8) in which *dmd*⁻/⁻ larvae exhibit prolonged survival due to protection by *fhl2b*, muscle injury does not occur as frequently or to the same extent and axons and neuromuscular junctions are less affected. Additionally, *fhl2b* improves macrophage response to myofiber injury, thereby accelerating recovery from myofiber damage. We

therefore propose that Fhl2 based therapy has the potential to be utilized to alleviate muscular dystrophy in body musculature.

Fhl2 is a co-transcription factor and can translocate to the nucleus to co-activate gene expression[50] and has been shown to be expressed in myogenic precursor cells to promote differentiation[44] and in Pax7 positive satellite cells during muscle regeneration[51,52]. However, we did not detect any Fhl2 positive myonuclei in the EOMs, instead Fhl2 was present mainly on the Z-discs of myofibers and likely executes its function there, similar to what has been observed in rat myocardiocytes[53]. *fhl2* is not widely expressed in muscle derived from the paraxial mesoderm in zebrafish between mid somitogenesis to 5 dpf[46] and its knockout did not have obvious consequences on muscle development in our study, and others[39,40], arguing against a primary role in skeletal muscle development. It has been shown that Fhl2 inhibits β-adrenergic stimulated calcineurin/NFAT signaling[53]. However, we did not find any DEGs known to be involved in this signaling cascade in *desma*⁻/⁻;*desmb*⁻/⁻ EOMs In rodent cardiomyocytes, Fhl2 overexpression has also been shown to inhibit SRF/ERK pathway activity[54] and block ERK2 translocation to the nucleus and subsequent transcriptional activity, offering protection from hypertrophy[55]. The WNT/β-catenin signaling pathway has previously also been linked to both Fhl2 and hypertrophy[43,44,56,57]. Fhl2 can act both to repress β-catenin mediated transcription in muscle as well as to co-activate gene transcription in kidney and colon cell lines, showcasing Fhl2's diverse functions depending on cellular context. In this study, knockout of *fhl2* genes led to increased expression of *ctnnb2* (β-catenin), indicating a repressive function of *fhl2* on *ctnnb2* in our mutants. Furthermore, knockout of *fhl2* genes also affected the expression of *wnt5a*, *wnt5b* and *wnt11*, in line with previous studies[58], suggesting a role for Fhl2 in both canonical and non-canonical WNT-signaling in zebrafish, which may explain the hypertrophic changes observed in EOMs lacking Fhl2. As zebrafish hearts lacking Fhl2 were not hypertrophic, and hypertrophy in mice hearts lacking Fhl2 show increased calcineurin/NFAT signaling[53,54], it is therefore possible that the function of Fhl2 and the process of hypertrophy differs in EOMs compared to cardiac muscles.

Fhl2 has been linked to metabolic enzyme targeting to the N2B region of titin, suggesting that it acts as an adaptor protein for high energy demanding regions within the sarcomere[59]. The N2B region has also been proposed to be an anchoring hub for mechanotransduction[60] and Fhl2 has previously been shown to be important for mechanotransduction in several scenarios where it is situated on F-actin and can translocate to the nucleus upon a change of tension to regulate gene expression[50,61]. Interestingly, *zyx*, *pdlim2* and *ldb3a*, identified in our transcriptional profiling of dystrophic EOMs, are also linked to mechanotransduction[62–64], introducing mechanosensing as a potentially important mechanism for EOM specific resistance to muscular dystrophy. This notion is strengthened by the fact that we found Fhl2 to be more abundant in a number of muscular dystrophy models, suggesting it is part of a general mechanism of defense in the EOMs. Knockout of *fhl2* genes leads to increased levels

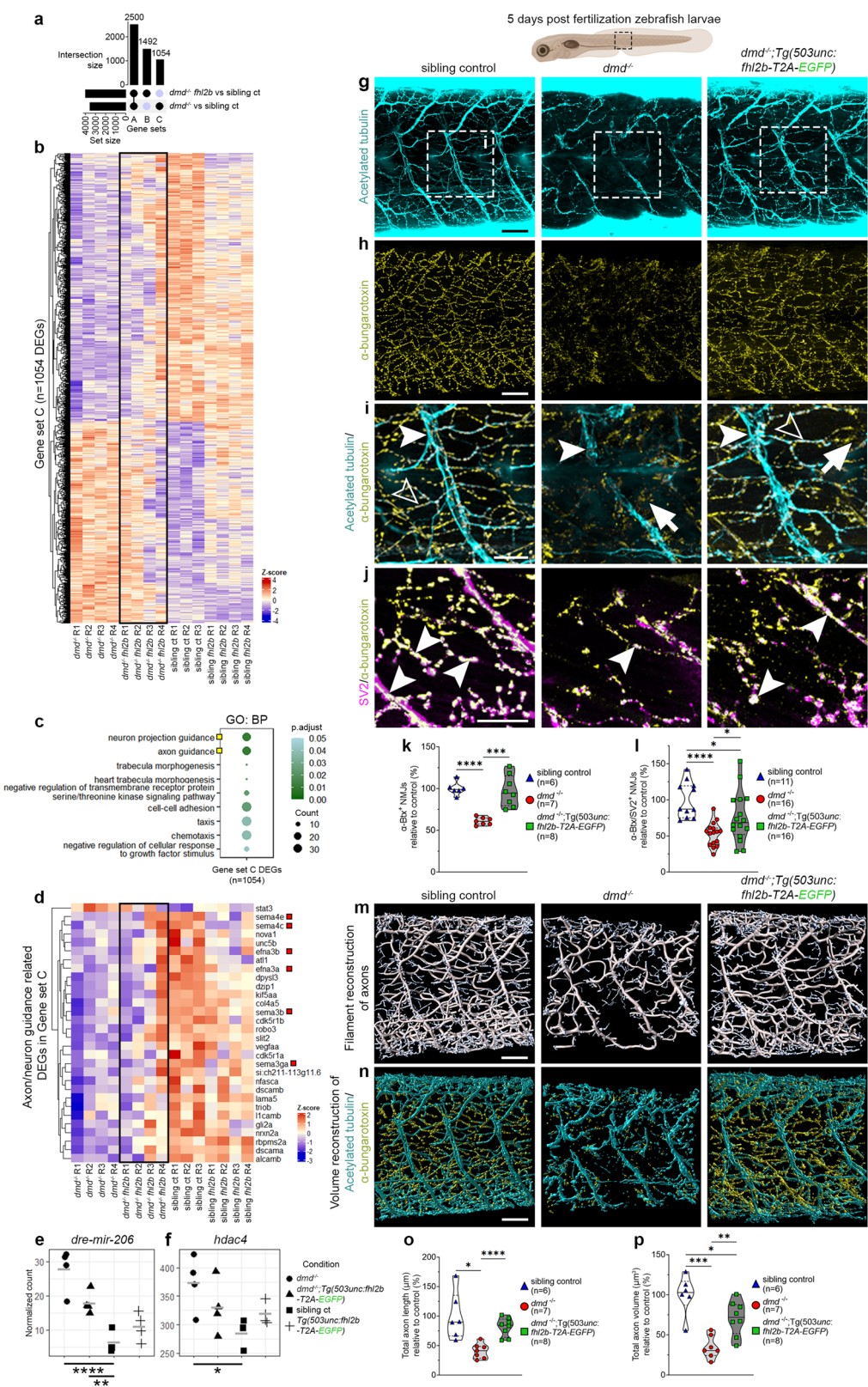

of cellular turn around and hypertrophy, emphasizing the importance of Fhl2 in EOM myofiber homeostasis, similar to what has been shown in rodent cardiomyocytes[39,55]. There was no redundancy between Fhl family members, however, we cannot exclude that the downregulation of *fhl1* genes in the quadruple mutant larvae might contribute to the cellular turnaround phenotype in the EOMs. We show that ectopic *fhl2b* expression in *dmd*[−/−] larvae leads to protection from myofiber

integrity loss. However, whether ectopic *fhl2b* has the same function in trunk muscle, EOMs and cardiomyocytes remains to be determined.

Duchenne muscular dystrophy models have shown alterations in NMJ patterning[65], non-surprising considering that the dystrophin-glycoprotein-complex (DGC) show accumulation at the NMJs in healthy muscle. Interestingly, our RNA-sequencing and in vivo data clearly demonstrate preserved expression of several genes related to axon

**Fig. 5 | Muscle specific overexpression of *fhl2b* ameliorates axon and neuromuscular junction integrity in *dmd*⁻/⁻ zebrafish larvae. a** Upset plot showing intersection of DEGs between *dmd*⁻/⁻:*Tg(SO3unc:fhl2b-T2A-EGFP)* vs sibling controls (*dmd*⁺/⁺, *dmd*⁺/⁻) and *dmd*⁻/⁻ vs sibling controls (Gene set A: disease-related DEGs shared between *dmd*⁻/⁻ and *dmd*⁻/⁻;*Tg(SO3unc:fhl2b-T2A-EGFP)*; Gene set B: DEGs unique to *dmd*⁻/⁻;*Tg(SO3unc: fhl2b-T2A-EGFP)* larvae; Gene set C: disease-related DEGs specific to *dmd*⁻/⁻ larvae). **b** Heatmap displaying the expression of 1054 DEGs in Gene set C. **c** GO terms enriched in genes from gene set C. Yellow boxes: GO terms related to axon and neuron projection guidance; BP: Biological Process. **d** Heatmap displaying the expression of axon and neuron guidance related DEGs, red boxes: *semaphorin* and *ephrin* DEGs. Normalized counts for **e** *dre-mir-206* (p.adj=0.00005, p.adj=0.009) and **f** *hdac4* (p.adj=0.02). Comparisons were made between *dmd*⁻/⁻ vs sibling controls and *dmd*⁻/⁻:*Tg(SO3unc:fhl2b-T2A-EGFP)* vs sibling controls, respectively. **g** Sibling control, *dmd*⁻/⁻ and *dmd*⁻/⁻;*Tg(SO3unc:fhl2b-T2A-EGFP)* larvae immunolabeled for acetylated tubulin and **h** α-bungarotoxin. **i** Magnification of dashed boxes in g), arrowheads: axons, open arrowheads: axon/NMJ overlap and arrows: NMJs lacking axon overlap. **j** Sibling control, *dmd*⁻/⁻ and *dmd*⁻/⁻;*Tg(SO3unc:fhl2b-T2A-EGFP)* larvae immunolabeled for SV2 and α-bungarotoxin. Arrowheads: SV2⁺/α-bungarotoxin⁺ NMJs. **k** Quantifications of α-bungarotoxin⁺ NMJs where $p < 0.0001$ for *dmd*⁻/⁻ vs sibling control and $p = 0.0008$ for *dmd*⁻/⁻ vs *dmd*⁻/⁻:*Tg(SO3unc:fhl2b-T2A-EGFP)*. **l** Quantifications of SV2⁺/α-bungarotoxin⁺ NMJs where $p = 4.6e^{-5}$ for *dmd*⁻/⁻ vs sibling control, $p = 0.044$ for *dmd*⁻/⁻:*Tg(SO3unc:fhl2b-T2A-EGFP)* vs sibling control and $p = 0.024$ for *dmd*⁻/⁻ vs *dmd*⁻/⁻:*Tg(SO3unc:fhl2b-T2A-EGFP)*. **m** Filament reconstruction of acetylated tubulin labeled sibling control, *dmd*⁻/⁻ and *dmd*⁻/⁻:*Tg(SO3unc:fhl2b-T2A-EGFP)* larvae. n) Volume reconstruction of acetylated tubulin/α-bungarotoxin labeled sibling control, *dmd*⁻/⁻ and *dmd*⁻/⁻:*Tg(SO3unc:fhl2b-T2A-EGFP)*. **o** Quantification of total axon filament length based on reconstructions in (**m**) where $p = 0.014$ for *dmd*⁻/⁻ vs sibling control and $p = 9.7e^{-5}$ for *dmd*⁻/⁻ vs *dmd*⁻/⁻:*Tg(SO3unc:fhl2b-T2A-EGFP)*. **p** Quantification of total axon volume based on reconstructions in (**n**) where $p = 0.0005$ for *dmd*⁻/⁻ vs sibling control, $p = 0.043$ for *dmd*⁻/⁻:*Tg(SO3unc:fhl2b-T2A-EGFP)* vs sibling control and $p = 0.002$ for *dmd*⁻/⁻ vs *dmd*⁻/⁻:*Tg(SO3unc:fhl2b-T2A-EGFP)*. Statistical analysis: Two-sided t-tests with Welch correction. Trunk region viewed is indicated in illustration above. Scale bar in **g, h, m, n**: 50 μm, **i, j**: 25 μm. Schematic images were adapted from https://www.biorender.com.

and motor neuron guidance and a clear improvement in axon and NMJ structure in *dmd*⁻/⁻;*Tg(SO3unc:fhl2b-T2A-EGFP)* compared to *dmd*⁻/⁻ larvae. *fhl2b* overexpressing larvae notably showed higher levels of axon guidance cues such as semaphorins, one of the major families of axon guidance molecules[66]. While muscle-secreted molecules have been shown to affect motor neurons directly[67], it is difficult to assess whether our results are a direct effect of *fhl2b* overexpression or a consequence of preserved myofiber integrity. Denervated myofibers undergo degeneration[68], likely adding to the overall muscle phenotype observed in *dmd*⁻/⁻ larvae. Previous studies have shown the beneficial effects of electrical muscle stimulation on DMD-deficient myofibers[69]. In our study, *fhl2b* overexpressing *dmd*⁻/⁻ larvae showed improved motor function, which is likely a consequence of both improved muscle and axon/NMJ structure. The DGC is abundant at cell adhesion sites, including the NMJ[70]. In the absence of a functioning DGC in our DMD models, the second major costameric stabilization unit, the laminin-integrin-talin complex, likely becomes more important. Fhl2 has been shown to bind several integrins in yeast and human HEK293 cells (integrin-α3A, α3B and α7A) and colocalize with integrins in myoblasts and mouse cardiac muscle at focal adhesion sites and the sarcolemma (integrin-α7β1)[71,72]. Furthermore, lack of integrin-α3 causes nerve terminal detachment in mouse NMJs[73]. We speculate that Fhl2 may contribute towards stabilizing the sarcolemma, and in extension the NMJs, potentially through interaction with integrins.

Fhl2 is linked to wound healing in several tissues, including muscle[51,74,75], and lack of Fhl2 or its suppression in these processes is coupled with chronical inflammation or deteriorating wound healing[76,77]. In muscle, Fhl2 has been suggested to regulate *il-6* and *il-8* production via MAPK signaling, playing a role in post-injury inflammation[78,79]. Our data show increased *il-8* levels under uninjured and *dmd*⁻/⁻ *fhl2b* overexpression conditions, supporting these claims. Recently, macrophages have been shown to be critical in muscle injury regeneration by inducing satellite cell proliferation[49]. Interestingly, our RNA-sequencing data show upregulation of macrophage-specific genes (*mmp13a, mpeg1.2*) in *Tg(SO3unc:fhl2b-T2A-EGFP)* and *dmd*⁻/⁻;*Tg(SO3unc:fhl2b-T2A-EGFP)* larvae. Furthermore, in needle-stick injured *Tg(SO3unc:fhl2b-T2A-EGFP)* larvae, macrophages are significantly more abundant in the early stage of myofiber repair and regeneration compared to controls. Subsequently, the number of Pax7 positive satellite cells is significantly higher at an early stage of regeneration. As a result, myofibers are restored more rapidly at the wound site. In summary, we propose that overexpression of *fhl2b* results in accelerated muscle regeneration via enhanced macrophage recruitment.

Taking advantage of the EOMs innate resistance to muscular dystrophies combined with genetic models constitutes a novel approach to identify treatment strategies for these devastating conditions. Our study shows Fhl2 upregulation in EOMs of several disease models, suggesting that Fhl2 is a strong candidate for the development of future therapeutic strategies for a common management of muscular dystrophies.

## Methods

### Animal husbandry and ethical approval

All experiments were performed in compliance with national and institutional laws and guidelines and the study is reported in accordance with ARRIVE guidelines. All zebrafish animal experiments were ethically approved by the Regional Ethics Committee at the court of Appeal of Northern Norrlands Umeå djurförsöksetiska nämnd, Dnr: A6 2020. The Mouse experiments were approved by the Animal Review Board at the Court of Appeal of Northern Norrland in Umeå Dnr: A22−2023. Human samples were collected with the approval of the Regional Ethical Review Board in Umeå (Dnr: 2010-373-31 M), in accordance with the principles of the Declaration of Helsinki.

Zebrafish larvae (*Danio rerio*) and adult fish were maintained from AB WT. Mutant lines used were *desma*ᵘᵐᵘ¹⁰, *desmb*ᵘᵐᵘ¹¹, *obscnb*ᵘᵐᵘ¹⁶, *plecb*ᵘᵐᵘ²⁵, *fhl2a*ᵘᵐᵘ³² , *fhl2b*ᵘᵐᵘ³³ and *sapje*ᵗ²²²ᵃ (referred to as *dmd*⁺/⁻). Transgenic lines used were *Tg(mylz2:EGFP)*ⁱ¹³⁵, *Tg(smyhc1:tdTomato)*ⁱ²⁶¹, *Tg(SO3unc:EGFP)*ᵘᵐᵘ³⁷ and *Tg(SO3unc:fhl2b-T2A-EGFP)*ᵘᵐᵘ³⁴. All zebrafish of the same genotype were reared from the same parental couple, to minimize genetic background bias across age. Additionally, WT zebrafish used originated from the same line utilized when generating the *desma*⁻/⁻;*desmb*⁻/⁻ mutant. Zebrafish were maintained by standard procedures on a 10/14 h dark/light cycle at 28 °C at the Umeå University Zebrafish Facility and fed twice daily (Live artemia cysts 10309, ZEBCARE B.V, Nederweert, The Netherlands). Leftover mouse tissue from 11 mice that were used in terminal experiments performed by other researchers was kindly donated (Leif Carlsson, Umeå University) and were of mixed WT background (western blot, eight weeks old: 129/Sv:CBA/J:C57BL/6 J:DBA2/J), immunohistochemistry, four weeks old: 129/Sv:CBA/J:C57BL/6 J:DBA2/J). Mice were maintained under 12:12 h light/dark cycles at constant temperature and humidity (22 °C and 50% humidity, fed formula 1310 breeding diet *ad libitum* (Altromin Spezialfutter GmbH, Lage, Germany)) and were euthanized by cervical dislocation. A total of ten EOM muscle samples were obtained at autopsy from five human donors (four men and one woman, ages 47−80) who, when alive, had consented to donate their eyes and other tissues post-mortem for transplantation and research purposes, according to Swedish law. There was no previously known neuromuscular disease among the donors. Schematic images were adapted from https://www.biorender.com under a subscription license.

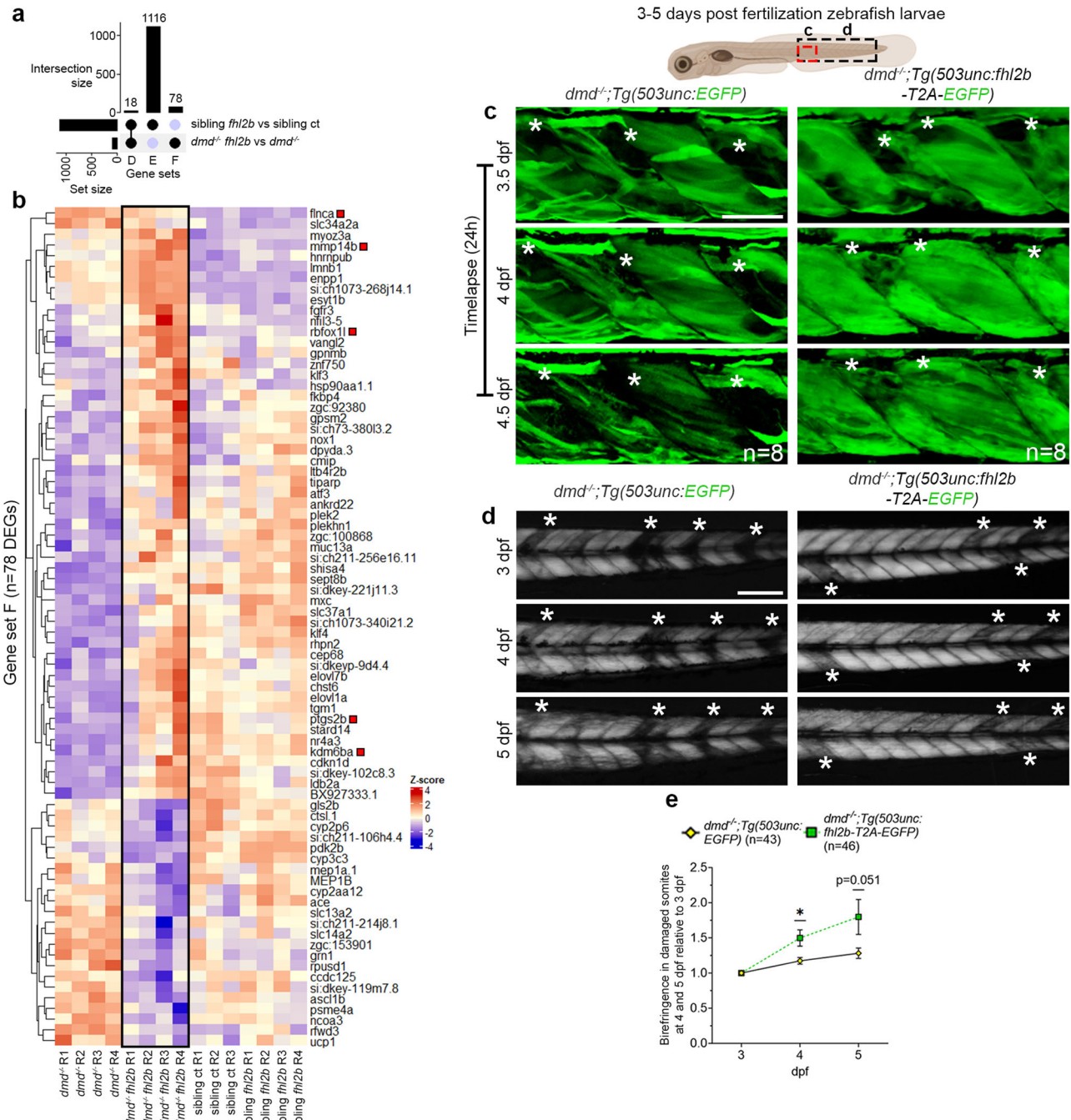

**Fig. 6 | *fhl2b* overexpression leads to upregulation of regenerative markers in *dmd*−/− zebrafish larvae. a** Upset plot showing the intersection of DEGs between *Tg(503unc:fhl2b-T2A-EGFP)* vs sibling controls and *dmd*−/−:*Tg(503unc:fhl2b-T2A-EGFP)* vs *dmd*−/− larvae (Gene set D: DEGs caused by *fhl2b* overexpression in both comparisons; Gene set E: DEGs caused by *fhl2b* overexpression in healthy sibling control conditions alone; Gene set F: DEGs caused by *fhl2b* overexpression in the *dmd*−/− disease condition). **b** Heatmap displaying the expression of DEGs in Gene set F. Red boxes indicate DEGs related to muscle regeneration. **c** Timelapse images of *dmd*−/−;*Tg(503unc:EGFP)* and *dmd*−/−;*Tg(503unc:fhl2b-T2A-EGFP)* larvae at 3.5, 4 and 4.5 dpf, respectively. Asterisks (*) indicate damaged somites. Area viewed is

indicated in zebrafish illustration above. **d** Birefringence images following the same *dmd*−/−;*Tg(503unc:EGFP)* and *dmd*−/−;*Tg(503unc:fhl2b-T2A-EGFP)* larvae at 3, 4 and 5 dpf. Asterisks (*) indicate damaged somites. Trunk region viewed is indicated in zebrafish illustration above. **E** Quantification of birefringence in damaged somites over time. Birefringence at 4 and 5 dpf was compared to 3 dpf in order to evaluate muscle regeneration progression over time. *p* = 0.012 at 4 dpf and *p* = 0.051 at 5 dpf. Statistical analysis in e, f: Two-sided Wald test with B/H-correction, **k, l, o, p:** Two-sided t-test with Welsh correction. Scale bar in **c**: 50 μm and d: 200 μm. Schematic images were adapted from https://www.biorender.com.

## Generation of *desma, desmb, fhl2a* and *fhl2b* mutant zebrafish using CRISPR/Cas9

*desma* and *desmb* zebrafish mutants were generated using methods previously described[80]. For *desma*, and *desmb* a guide RNA (gRNA) targeting exon 1 was synthesized using the sequences ATT-CAGCCTCCGCCGAGTCGG and GGTGGGTCGGGCAGCTCTCGG

respectively, and was coupled with a gRNA scaffold[80]. The gRNA was then transcribed using the MegaShortScript T7 (Invitrogen) kit and co-injected with Cas9 protein (New England Biolabs) into one-cell stage zebrafish eggs. Injected larvae were grown to adulthood, outcrossed into WT zebrafish, and screened to identify founders containing germline mutations. Mutant zebrafish larvae carrying a 5 bp and a

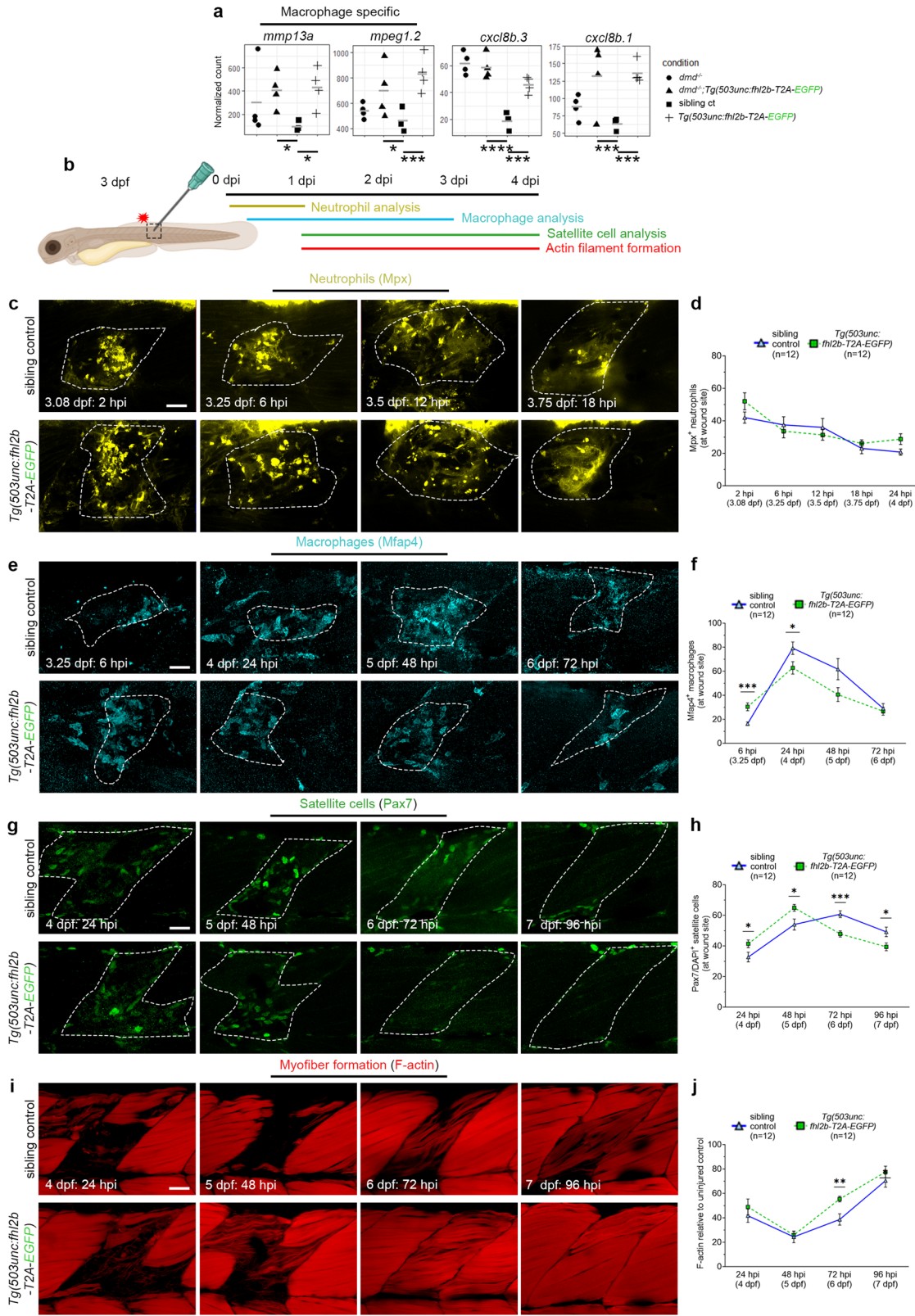

20 bp deletion in the *desma* and *desmb* genes, respectively, were chosen for further examination. Genotyping of *desma* mutant larvae was accomplished by standard PCR protocols using forward 5′-ATAG AAGTGGGCGCCAATG-3′ and reverse 5′-GTCTTGAGGAGCCAGAG GAA-3′ primers and *desmb* mutant larvae were genotyped using forward 5′-AGCCACTCTTATGCCACCTC-3′ and reverse 5′GCGGTCA TTTAGATGCTGAAG-3′ primers. The PCR products were then digested overnight using HinfI and AluI for *desma* and *desmb*, respectively, and analyzed on a 2% agarose gel. *fhl2a* and *fhl2b* mutants were generated as described above, using gRNA sequences AAGAAGTATGTCC TGCGTGAGG and CGGGAAGAAGTACGTCCTGCGG respectively. To genotype *fhl2a* mutants forward 5′ATAGAAGTGGGCGCCAATG-3′ and reverse 5′TGGGTTTCTTGCATTCCTCG-3′ primers were used and the resulting PCR product was digested with EcoNI to screen for

**Fig. 7 | Muscle wound healing is enhanced in *fhl2b* overexpressing larvae.**
**a** Normalized counts for *mmp13a* (p.adj=0.012, p.adj=0.023), *mpeg1.2* (p.adj=0.015, p.adj=0.0009), *cxcl8b.3* (p.adj=7.4e⁻⁸, p.adj=0.0007) and *cxcl8b.1* (p.adj=0.0005, p.adj=0.0001). Comparisons were made between *dmd⁻/⁺:Tg(503unc: fhl2b-T2A-EGFP) vs* sibling controls (*dmd⁺/⁺*, *dmd⁺/⁻*) and *Tg(503unc: fhl2b-T2A-EGFP) vs* sibling controls, respectively. **b** Wound healing assay experimental setup indicating area of needle-stick injury and timepoints for the different analyzes presented below.
**c** Lateral view of injured somites in sibling controls and *Tg(503unc:fhl2b-T2A-EGFP)* zebrafish embryos immunolabeled with neutrophil specific Mpx antibody between 2–18 hpi. Dashed lines indicate wounded areas. Quantifications of Mpx⁺ neutrophils at wound size is presented in **d. e** Lateral view of injured somites in sibling controls and *Tg(503unc:fhl2b-T2A-EGFP)* zebrafish embryos immunolabeled with macrophage specific antibody Mfap4 between 6–72 hpi. Dashed lines indicate wounded

areas. **f** Quantification of Mfap4⁺/DAPI⁺ cells in sibling controls and *Tg(503unc:fhl2b-T2A-EGFP)* at 6 (*p* = 0.0019), 24 (*p* = 0.0374), 48 and 72 hpi. **g** Lateral view of injured somites in sibling controls and *Tg(503unc:fhl2b-T2A-EGFP)* zebrafish embryos immunolabeled with satellite cell specific Pax7antibody at 24–96 hpi.
**h** Quantification of Pax7⁺/DAPI⁺ cells in sibling controls and *Tg(503unc:fhl2b-T2A-EGFP)* at 24 (*p* = 0.0445), 48 (*p* = 0.0198), 72 (*p* = 0.0002) and 96 hpi (*p* = 0.0246).
**i** Lateral view of injured somites in sibling controls and *Tg(503unc:fhl2b-T2A-EGFP)* zebrafish embryos labeled with phalloidin (F-actin) at 24–96 hpi. **j** Quantification of F-actin intensity at wound site in sibling controls and *Tg(503unc:fhl2b-T2A-EGFP)* at 24, 48, 72 (*p* = 0.0045) and 96 hpi. Statistical analysis in a: Two-sided Wald test with B/H-correction, **d**, **f**, **h**, **j**: Two-sided t-tests with Welch correction. Data in graphs is presented as mean ± SEM. Scale bar: 25 µm. Schematic images were adapted from https://www.biorender.com.

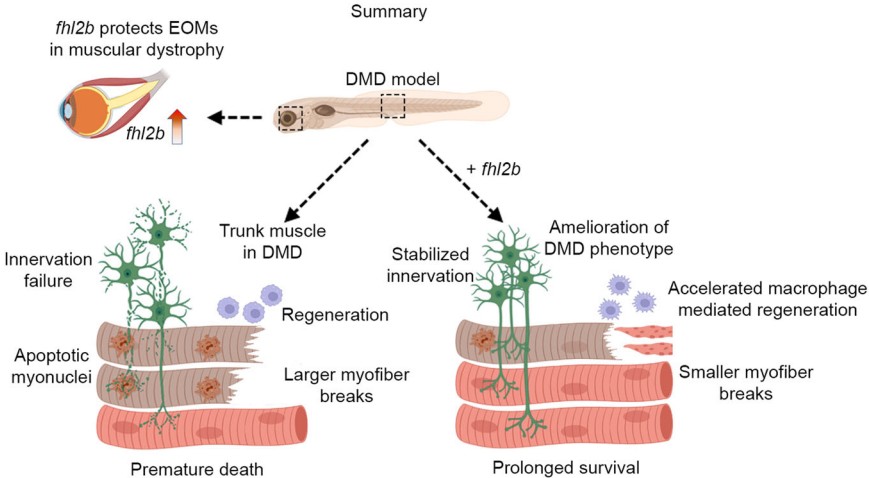

**Fig. 8 | Model for *fhl2b* mediated rescue of muscular dystrophy.** *fhl2b* is upregulated and protects EOMs from muscular dystrophy conditions. *dmd⁻/⁻* larvae overexpressing *fhl2b* survive for extended amounts of time due to improved

muscle integrity with fewer myofiber detachments and breaks, improved axon and NMJ stability, and accelerated myofiber regeneration. Schematic images were adapted from https://www.biorender.com.

mutations. *fhl2b* mutants were genotyped using 5'CCCTTTCAACCTGTCTGCAC-3' and reverse 5'GCAGGTATTGGAGTAGAGGCT-3' primers and the PCR product was digested using HpyCH4IV. The resulting *fhl2a* and *fhl2b* mutants chosen for investigation both carried 7 bp deletions in exon 1.

### Transgenesis
An overexpression vector (*503unc:fhl2b-T2A-EGFP*) containing the muscle-specific promoter 503unc driving expression of *fhl2b* coupled to EGFP via T2A was acquired from Vectorbuilder (Neu-Isenburg, Germany). Subsequently, AgeI and NheI were used to remove the *fhl2b* cassette and fused using ligase, to generate a *503unc:EGFP* vector. Both vectors were co-injected in one cell-stage AB WT zebrafish eggs with Tol2-transposase RNA, at 30 ng/ul, respectively. Founders were screened using EGFP expression detected under a fluorescent dissecting microscopy and out crossed with wild type fish to generate stable lines used in our experiments.

### Immunohistochemistry and TUNEL assay
The muscle samples from both humans and mouse were immediately mounted on cardboard after collection and rapidly frozen in propane chilled with liquid nitrogen and then stored at −80 °C until sectioned. Serial transverse sections, 5–8 µm thick, were cut using a cryostat (Reichert Jung; Leica, Heidelberg, Germany) at a temperature of −23 °C and collected on glass slides which were processed for immunostaining as described previously[18]. Adult zebrafish at the appropriate age were deeply anesthetized with 0.01% ethyl 3-aminobenzoate methane sulfonate (Tricaine, MS-222, Sigma Aldrich) and sacrificed by decapitation. The heads and posterior trunks were fixed in 2%

paraformaldehyde (PFA) for 1 h at RT and stepwise incubated in 10, 20 and 30% sucrose for 12 h each, at 4 °C. The heads and trunks were then mounted separately on cardboard using OCT cryomount and snap frozen in liquid nitrogen chilled propane, and finally serially cut into 12 µm (trunk tissue) or 14 µm (head tissue) thick sections using a cryostat (Reichert Jung; Leica, Heidelberg, Germany). Trunk preparations were cut from adult zebrafish at the anal opening and cut again 8–10 mm caudally to the initial cut. Sections were always made from the proximal end of the specimen to assure the highest possible section similarity between fish. Sectioned zebrafish muscle tissue was rinsed in phosphate buffered saline (PBS) for 15 min before the addition of blocking solution (1% blocking reagent (Roche Diagnostics GmBH, Manheim, Germany) with 0.4% Triton X, 5% dimethyl sulfoxide (DMSO) and 0,1 % TWEEN20) for 1 h, at RT. Primary antibodies were diluted in blocking solution and were applied for 48 h, at 4 °C. Sections were then washed 3 times in PBS and incubated with secondary antibodies for 24 h at 4 °C, washed 3 times in PBS and mounted using 80% glycerol. Adult 12 months old zebrafish hearts were dissected as previously described[81] and fixed for 1 h at RT in 2% PFA before stepwise incubation in sucrose as described above. Next, the hearts were orientated in the same position in OCT cryomount before they were carefully placed in −80 °C for storage until sectioned at 20 µm.

For whole-mount immunohistochemistry, larvae were fixed in 2% PFA for 1 h at RT, washed 3 times in PBS, acetone cracked for 1 h at −20 °C, washed 3 times in PBS and incubated for 1 h with blocking solution at RT. The blocking solution was replaced by a blocking solution containing the primary antibody and incubated 48–96 h depending on the stage. Larvae were washed 3 times 15 min in PBS before the addition of secondary antibodies diluted in blocking

solution and incubated 24–48 h depending on the stage before being washed 3 times 15 min in PBS again. Lastly, larvae were allowed to equilibrate to 80% glycerol for 1 h before being mounted on glass slides for imaging. DAPI and phalloidin were added with the secondary antibodies. All primary and secondary antibodies used are presented in Supplementary Data 4.

Additionally, a TUNEL assay (Click-iT, Alexa Flour, Invitrogen, Thermo Fisher Scientific) was performed to label nuclei containing degraded DNA molecules, following the manufacturer's description.

## Western blot

Protein extraction was performed in Pierce™ RIPA buffer (Thermo Fisher Scientific, 89901), supplemented with a protease and phosphatase inhibitor cocktail (Thermo Fisher Scientific, 1861282), employing a handheld tissue ruptor. Protein concentration was quantified using Pierce™ BCA Protein Assay Kit (Thermo Fisher Scientific, 23225). An equal amount of proteins were run in Any KD precast polyacrylamide gel (Bio-Rad Laboratories) and transferred onto polyvinylidene fluoride (PVDF) membrane. Primary antibodies used were as follows: FHL2 (Medical and Biological laboratory, K0055-3, 1:1000) for mouse muscle lysate, FHL2 (Atlas Antibodies, HPA005922, 1:1000) for human muscle lysate, and GAPDH (Abcam, AB8245, 1:2000). Anti-rabbit IgG, HRP-linked secondary antibody (Cell signaling, 7074, 1:2000) and anti-mouse IgG HRP-linked secondary antibody (Cell signaling, 7076, 1:2000) were used before membrane underwent incubation with SuperSignal West Pico PLUS Chemiluminescent Substrate (Thermo Fisher Scientific, 34580) and analyzed using an Odyssey Fc Dual-Mode Imaging System (LI-COR Biotechnology). Uncropped blots, including all samples, are displayed in the Source data file for Fig. 3.

## Whole-mount in situ hybridization

RNA probes for in situ hybridization were synthesized using a PCR method for RNA probes as previously described[82]. Zebrafish larvae were fixed in 4% paraformaldehyde (PFA) overnight at 4 °C and dehydrated in 30%, 50%, 70% and 100% methanol and stored in 100% methanol at −20 °C until use. Whole-mount in situ hybridization was performed as described previously[83]. The RNA probes were based on the sequences of *fhl2a* (NM_001003732.1) and *fhl2b* (NM-001006028.2). Primers used to generate probes were: *fhl2a* forward 5′-CCTGCGTGAGGACAACCCATAC-3′ and reverse 5′-GGTCTCATGCC AGCTGTTTCC-3′. *fhl2b* forward 5′-GCAAAAAGCCCATTGGCTGC-3′ and reverse 5′-CAGGTCTCATGCCAGCTGTTC-3′.

## BrdU treatment

To label the proliferating cell population in our model, zebrafish larvae were collected at 4 dpf treated with BrdU at a concentration of 10 mM and incubated overnight at 28.5 °C. 5 dpf larvae were fixed in 4% PFA for 2 h at room temperature. BrdU was counter-stained with anti BrdU-555 (1:500).

## Wound assay

3 dpf larvae were anesthetized in tricane in embryo medium and placed dorsal side up on a 2% agarose gel in a petri dish. Mechanical injuries were targeted to the somites on the dorsal side of embryos, above the cloaca, using a single stab with a sharpened glass capillary. This generated extensive muscle tissue damage localized to approximately one somite. The larvae were then washed in fresh embryo medium and reared until the appropriate stage. More than 95% of larvae survived this procedure.

## Imaging and automated measurements

Imaging of sectioned tissue and whole-mount larvae was performed using a Nikon A1 confocal microscope (Nikon, Tokyo, Japan). Images are presented as five merged Z-stacks of representative areas or Z-depth in all figures. Automated measurements of EOM myofiber

areas was performed using CellProfiler[84]. Pre-photographed confocal images of sibling control, *dmd^{−/−}* and *dmd^{−/−};Tg(S03unc:fhl2b-T2A-EGFP)* 5 dpf larvae immunolabeled for acetylated tubulin and α-bungarotoxin were uploaded to the Imaris Arena (10.1, Bitplane, UK) for quantitative measurements of axons and NMJs. Each image was quantified on four somites directly dorsal to the larval cloaca. To assess acetylated tubulin lengths and volume, images were first individually segmented in the Labkit extension of ImageJ[85] to remove background signal. A newly generated layer containing a robust signal was passed back to Imaris Arena. Next, the FilamentTracer tool was utilized to generate 3D filament structures, to quantify axon length (μm). Settings utilized were: Algorithm – Autopath (no loops), shortest distance from distance map, Threshold – Auto. To quantify axon volume (μm³), the Surface tool was applied to the same layer using the following settings: Threshold – Absolute intensity, filter – auto. To quantify normal sized NMJs, the surface tool was applied to the α-bungarotoxin layer using the following settings: Threshold – Absolute intensity. A voxel filter excluding surfaces <37 voxels was applied to all images, determined by the smallest sibling control NMJs detected in 5 samples.

## RNA-sequencing and analysis

For each group, 30 five months old and size matched (24 mm ± 3 mm) adult *desma^{−/−};desmb^{−/−};mylz2*:EGFP mutants and WT^{AB};*mylz2*:EGFP were euthanized by Tricaine. These were then swiftly subjected to EOM dissection, essentially as described[20] but without paraformaldehyde fixation. All six EOMs still attached to the sclera were transferred into RNA-Later solution kept on ice and the EOMs were further cleaned, pooled and frozen in RNA-later. A piece of the lateral trunk including the slow domain myofibers from each fish was also excised and treated in the same manner. For EOM and trunk sequencing of 20 months old zebrafish, the same process was applied except the size-matched animals were 31 mm ± 3 mm and each group contained 12 animals. In total, 254 fish were harvested. For 5 dpf zebrafish larvae, four groups of 10 larvae each containing sibling controls, *Tg(S03unc:fhl2b-T2A-EGFP)*, *dmd^{−/−}* and *dmd^{−/−};Tg(S03unc:fhl2b-T2A-EGFP)* larvae were cut diagonally distal to the swim bladder to exclude the majority of the gastrointestinal region and the head from the analysis. These larvae were then stored in RNA-later at −80 °C until further use.

Pooled tissue was thawed and briefly rinsed in PBS before RNA extraction was performed using the TRIzol (Invitrogen, Thermo Fisher Scientific) reagent standard procedures and isolated in 15 μl of RNAse free H₂O. Quality controls were performed using a bioanalyzer and a RIN value greater than 8 was considered acceptable for analysis. Library preparation was performed using the TruSeq Stranded mRNA Library Prep kit (cat#20020595, Illumina, Inc.), including poly-A selection, according to the manufacturer's instructions. Unique index adapters were used (cat#20022371, Illumina, Inc. 15 cycles of amplification). The RNA-seq libraries were sequenced on a Novaseq 6000 Sequencing system (Illumina, Inc.) obtaining in average ~78.5 million 150 paired end reads per library. Library preparation and sequencing was performed at SciLifeLab Stockholm.

Fastq files were quality controlled using FastQC (https://www.bioinformatics.babraham.ac.uk/projects/fastqc/) and raw reads were mapped to the zebrafish genome (GRCz11) using STAR (2.7.6a)[86]. Normalization and differential expression analysis were performed using DESeq2[87], only genes with 10 or more reads were processed. Differentially expressed genes (DEGs, padj = <0.05) from different comparisons were intersected using the UpSet plot function from the ComplexHeatmap package[88] and Gene Ontology (GO) analysis was performed using the clusterProfiler package[89] (padj = <0.05) and plotted using the enrichplot package. Expression of single genes was plotted using ggplot. All statistical analysis related to RNA-sequencing data was performed within the R environment (version 4.2.3) using basic built-in functions and publicly available packages listed above.

These are open-source tools and can be accessed via Bioconductor (hppts://Bioconductor.org/).

## qPCR

Quantitative PCR was performed on using the same RNA-extraction method as described above on whole 5 dpf larvae. cDNA was synthesized using SuperScript IV (Invitrogen, Thermo Fisher Scientific). Primers used for all genes are presented in Supplementary Data 5. *β-actin* was used as a reference gene. The samples were run using an Applied Biosystems VIIA-7 Real Time PCR system (Thermo Fisher Scientific) using FastStart universal SYBR green master mix (Roche).

## Physiological properties of zebrafish muscle force/tension relationship

Zebrafish larvae were examined with length- force experiments as previously described by Dou et al[90]. and Li et. al.[91]. In brief, 5 and 6 dpf WT and *desma⁻/⁻;desmb⁻/⁻* larvae were euthanized and mounted with aluminum clips between a force transducer and a puller for length adjustment. The bath was perfused at 22°C in a MOPS buffered physiological solution with a pH of 7.4. The preparations were allowed to acclimatize in the solution for at least 10 min before initiating contractions using single twitch stimuli with 0.5 ms pulses at 2-min intervals and supramaximal voltage, via platinum electrodes placed on each side of the larvae. Measurements were initiated at slack length (Ls) and length was gradually increased by 10% steps every 4 min, between stimuli, until a decline in active force was observed. At each length, both active and passive force were recorded twice, and an average was calculated. The values were plotted against relative stretch (lambda=length/Ls).

## Spontaneous swimming and resistance swimming

5 dpf WT and *desma⁻/⁻;desmb⁻/⁻* zebrafish larvae were placed in a 48 well-plate inside the Viewpoint ZebraBox system (Viewpoint Behavior Technology) to determine spontaneous movement patterns. Larvae were carefully touched by the tail to ensure mobility and were subsequently allowed to acclimatize to the environment for 15 min before a 60 min protocol with a steady light was initiated. The movement thresholds were set to inactivity = 0 and large movements = 1. Inactivity, small movement and large movement counts, swimming distance and swimming duration were recorded. For resistance swimming, 4 dpf larvae were placed in 1% methyl cellulose in 1x E3 medium and reared at 28.5 °C overnight. The following day, larvae were analyzed using fluorescent microscopy.

## Statistical analysis

Myofiber and myonuclei counts were performed in the whole slow domain of adult zebrafish trunks except when F310 positive myofibers were counted outside of the slow domain. F310 positive myofibers were then counted in 2 × 0.6 mm squares (20x magnification) just medial to the intermediate fast domain. The myofibers of each fish was counted twice, once for each side of the fish for all quantifications except for F310, where a total of four squares were counted, two for each side of the fish. For cardiac dimension measurements, the major and minor axis of the heart were added together and divided by the size of individual fish before comparisons between genotypes. The myofibers and myonuclei in the EOMs were counted separately in a total of two per fish. The medial rectus muscle was used. In embryonic experiments including Pax7, BrdU or TUNEL positive cells in the trunk muscle, somite numbers 13–22 was counted and the total number of positive cells were divided by the numbers of somites. For quantifications of α-bungarotoxin positive NMJs somite number 17–20 were counted. For quantifications of SV2/ α-bungarotoxin double positive NMJs, somites 18-19 were counted. Acetylated tubulin volume and length were measured for somites 17–20. For broken myofiber counts, somites 8–24 were counted. The number of larvae in the experiments are presented in the graphs. Measurements were taken from distinct samples and not from repeated measurements of the same samples. Normality tests were performed prior to the statistical analysis for each experiment.

All data was collected in Microsoft Excel and plotted in GraphPad Prism 10.0. Statistical analysis was performed using two-sided t-tests with Welch correction, $p = <0.05$ was considered significant (*$p < 0.05$, **$p < 0.005$, ***$p < 0.0005$, ****$p < 0.0001$). All data are presented as mean ± standard error of mean (SEM). Kaplan-Meier log rank test was used to determine the difference between genotypes in the survival analysis. $p = <0.05$ was considered significant.

## Reporting summary

Further information on research design is available in the Nature Portfolio Reporting Summary linked to this article.

## Data availability

The RNA-sequencing data generated in this study have been deposited in the Gene Expression Omnibus (GEO) database under accession code GSE242137. The processed RNA-sequencing data generated in this study are provided in the Supplementary Data 1, 2, and 3. Source data are provided with this paper.

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

## Acknowledgements

The authors acknowledge support from the National Genomics Infrastructure in Stockholm and Uppsala funded by Science for Life Laboratory, the Knut and Alice Wallenberg Foundation and the Swedish Research Council, and SNIC/Uppsala Multidisciplinary Center for Advanced Computational Science for assistance with massively parallel sequencing and access to the UPPMAX computational infrastructure. The computations were enabled by resources in projects SNIC 2022/22-442 and NAISS 2023/22-287 provided by the Swedish National Infrastructure for Computing (SNIC) and the National Academic Infrastructure for Supercomputing in Sweden (NAISS) at UPPMAX. The authors acknowledge Philip W. Ingham for generously sharing the *Tg(smyhc1:tdTomato)*[i261] and *Tg(mylz2:EGFP)*[i135] zebrafish lines. The authors thank Leif Carlsson for the donation of mouse tissue. AA was funded by grants from Hans-Gabriel and Alice Trolle-Wachtmeister's Foundation for Medical Research. JvH was funded by biotechnology grant for basic science FS 2.1.6-1911-22, the Medical Faculty, Umeå University. FPD was supported by research grants from the Swedish Research Council (Dnr 2018-02401), Västerbotten County Council (Central ALF and Spjutspetsmedel), Kronprinsessan Margaretas Arbetsnämnd för synskadade (Stiftelsen KMA), Ögonfonden. ND was supported by Kempestiftelserna, Ögonfonden and Arnerska forskningsfonden.

## Author contributions

N.D., Jv.H., and F.P.D. designed the study. Experiments were performed, analyzed, and interpreted by N.D., A.K., I.N., H.N., M.C., J.-X.L., J.L., A.A., and L.J.B. Transcriptional data was analyzed and interpreted by ND and IN, with supervision from S.R. N.D. wrote the manuscript with supervision and input from S.R., J.v.H. and F.P.D. N.D., Jv.H. and F.P.D. secured funding for the project.

## Funding

.

## Competing interests

The authors declare no competing interests.
