## [Peer Review File · Nature Communications]

REVIEWER COMMENTS

Reviewer #1 (Remarks to the Author):

The manuscript by Denhag and colleagues is focused on the role of fhl2b as having ameliorative properties in the zebrafish dystrophin-deficient model of Duchenne muscular dystrophy (DMD). Extraocular muscles (EOM) in DMD are spared from muscle wasting and dystrophic symptoms in patients. The authors compared differences in gene expression as a response to muscle dystrophies between the EOMs and trunk muscle in zebrafish via transcriptomic profiling. The authors demonstrate that Fhl2 is upregulated in response to knockout of desmin, plectin and obscurin, intermediate filament proteins whose knockout causes different muscle dystrophies, and contributes to DMD disease protection. Fhl2b ectopic overexpression can partially rescue dystrophic symptoms in the sapje/DMD model. The authors conclude that FHL2 overexpression can protect against dystrophic pathologies.

This is a very interesting study that integrates several model systems in zebrafish to identify a protective mechanism against muscular dystrophy that would be of high interest to the muscular dystrophy and general muscle biology fields. However, some key questions with regards to the extrapolation of findings from that of the extraocular muscles to that of fhl2b transgenic overexpression in dystrophic/dmd mutant zebrafish remain. Questions with regards to the mechanism of protection and FHL2's function need to be elucidated. Methodology is appropriate and statistical tests are appropriately used. The statistical approach and design are appropriate. This manuscript is well-written and with some elucidation of key experiments, would be a valuable addition to the muscular dystrophy field.

Overall Comments:

1. There's some overlapping expression and suggestive overlapping function between the FHL proteins. Does FHL1 and/or FHL3 increase in expression in the fhl2a, fhl2b and/or double fhl2a/b knockout zebrafish?
2. The authors mention the lack of publications demonstrating FHL2 is expressed extraocular muscles. Have the authors compared the transcriptomes from the fhl2b EOMs fish to that of published mouse studies demonstrating that EOM's are resistant to dystrophic pathologies (Pacheco-Pinedo et al., *Physiol. Genom.*, 2009)?

3. Line 204 “however, fh12b likely has an additional role in hypertrophic protection”. How exactly do the authors know that Fhl2b has an additional hyperprotective role? Can the authors explain why Fhl2a overexpression was not evaluated?

4. FHL2 has been shown to modify the beta-adrenergic stimulation response in hypertrophic hearts in mice (Kong et al., *Circulation*, 2001). The authors demonstrate here that Fhl2 causes EOM myofiber hypertrophy (Figure 3). Did the authors evaluate hypertrophy in the heart and/or skeletal muscle of their fh12b and/or fh12a/b double KO mutants?

5. Similar to my previous point, FHL2 has been shown to prevent pathological cardiac, and likely other tissue, growth (Hojayav et al., *Mol. Cell. Biol.* 2012) via a calcineurin-dependent pathway. Did the authors identify any calcineurin pathways that were affected in the overexpressed transgenic fish?

6. The biggest claim of innervation in response to Fhl2b transgenic overexpression needs a bit more follow-up. The authors show some bungarotoxin (Sup. Fig. S7; Figure 5E-G) staining but no quantification of this or demonstration of functional improvement as a result of increased innervation. The authors should really quantify the percentage of innervation in response to Fhl2b overexpression in the dmd mutants, and perhaps perform some additional acetylcholinesterase assessments of the transgenic muscles.

7. Another interesting finding, or lack thereof, by the authors is that there appears to be little effect on WNT signaling via Fhl2 overexpression if I am interpreting the transcriptomic data correctly? Did the authors directly probe WNT signaling in either the transgenic fh12b fish or knockout fish? FHL2 has been shown to be a direct interactor with beta-catenin to regulate WNT signaling and muscle differentiation. I would be curious to know if this is or is not the case in the authors' model(s).

Reviewer #2 (Remarks to the Author):

This manuscript first describes a new zebrafish dystrophy model followed by an investigation of the transcriptomics of extraocular versus trunk muscle (EOM being usually spared in dystrophy). This revealed that the four and a half LIM domain protein fh12 is upregulated in EOM in three different zebrafish dystrophies and suggests it may lead to the protection in EOM. Overexpression of fh12b in the most severe zebrafish dystrophy model improved their survival, suggesting Fhl2 as a potential target for treatment.

This research is extremely interesting, since members of the FHL family (especially FHL1 and FHL2) were shown previously to be involved in the stress response of skeletal muscle and heart during disease. Previous studies have demonstrated that at least in mouse FHL2 is NOT expressed in skeletal muscle (Lange et al., 2002) and only expressed in the heart and FHL3 was reckoned to be the main FHL in skeletal muscle (Fimia et al., 2000; Morgan and Madgwick, 1999). However, EOM are notorious for their peculiar status a bit inbetween cardiac and skeletal muscle, so the FHL2 expression in them may well also be the case in mouse/human - this should be shown by Western blot to provide more support for the potential translational value of this study. Unfortunately the images in Figure 2G and H that show FHL2 expression by immunofluorescence in human and mouse EOM are of too low quality to conclude much (signal in G extremely weak, no counterstain of any sort; why is the signal in the mouse in H suddenly only peripheral in the fibres when it appeared to be throughout the cytoplasm in fish and potentially human?). In addition, the expression levels of the zebrafish homologues of Fhl3 should be shown to account for upregulation/downregulation in the different experimental scenarios.

Specific comments:

1. Show Fhl2 expression in mouse and human EOM by Western blot.
2. Analyse fhl3 expression in the various models
3. FHL2 was shown in other system to be potentially involved in gene transcription. While the authors report that they did not see any signal for Fhl2 in the nucleus, its residence time there might be really brief and only detectable by inhibition of nuclear export pathways. The observation that it is actually the communication between muscle cells and nerves that is improved in the "rescue" fish is intriguing and I think necessitates a targeted investigation whether the Fhl2b fish have higher expression levels of miR-206 - which is well known to mediate communication between these two cell types.
4. Discuss Okamoto et al., 2012 FASEB & Friedrich et al., 2014; Basic Res Cardiol (FHL2 overexpression in cardiomyopathy); Hojayeve et al., 2012 (FHL2 & calcineurin)

Reviewer #3 (Remarks to the Author):

This manuscript examines the role of the gene *fhl2b* for its potential role in sparing the extraocular muscles in various forms of muscular dystrophy using a large number of zebrafish genetic mutants. This study is very well done, and the results are very exciting. The improved function of the body skeletal muscle after up-regulated expression of *fhlb2* is quite striking, and the data are important. I have only a few suggestions to improve readability and clarity, and all are relatively minor and straightforward. I also have made suggestions of literature that should be included in the Discussion section.

1. Please include supplemental figure 1 as a regular figure as well as supplemental figure 9. There is basic information in these figures, as they relate to the entire study, within these figures.

2. There are a number of studies in adult mammalian skeletal muscle showing that FHL2 is shown to be expressed in myogenic progenitor cells, and this literature should be discussed. FHL2 was shown to interact with FOXK1 and promoted proliferation of myogenic progenitor cell populations (Martin et al., J Cell Biol. 2002; Shi et al. Stem Cells 2010). This was shown in other studies of skeletal muscle satellite cells (Zhu et al., Genes 2022). Shi et al. (2010) also show that *fhl2* null mice have perturbed muscle regeneration specifically through cell cycle arrest, and that FHL2 expression correlated with expression of MyoD and Myogenin. These various results and what they mean in the context of your very interesting study need to be added to the Discussion, particularly where attributing other roles and locations for this protein (e.g. in the paragraph starting on line 331).

3. In addition, while you did not detect FHL2B-positive myonuclei in the EOMs, it may well be that precursor cells express it. This needs to be discussed, particularly in light of the Pax7-positive precursor cells you show that others were unable to demonstrate within the EOM of zebrafish (Saera-Vila et al., IOVS. 2015). It is hard to tell in Figure 6, for example, where the Pax7-expressing cells are located. In the controls, they seem to be within the connective tissue rather than associated with muscle fibers. While removal of debris is an important factor here, it is likely (based on the literature) that the muscle precursor cells are more active in the presence of elevated levels of FHL2B and more numerous in the EOMs.

4. There is also evidence that FHL2 interacts with muscle integrin receptors, which would play a role in mechanical stabilization of the muscle cells (Samson et al., 2004). This potential sparing mechanism also should be discussed in the Discussion section.

5. Because you are comparing multiple groups in almost all your graphs, a box plot gives a cleaner, more easily understandable summary of the data. The median is clearer. I suggest using box plots for displaying data summaries. They also will show clear outliers in the data.

6. The figures are rather small. The text is hard to read, even when I am reading this on my giant computer monitor. I suggest reducing the white space and enlarging the figures so that the details of the histology are clearer, as well as for the graphs, so that the labels can all be read.

7. Figure 5G is out of focus; please replace these photos. In 5G control, are each of the yellow dots a separate postsynaptic part of the neuromuscular junction? Are the dark spaces all single muscle fibers in cross section? It is quite difficult to assess what one is seeing. A crisper image and increased size will certainly help, but it might be useful to provide a photograph of a bright field image.

Other

8. Figure title for Figure 2 would be more accurate as “fhl2b is upregulated in the EOM in response to desmin-related muscular dystrophy”.

9. It would be good to temper line 329 to say that fhl2b therapy has the potential to alleviate muscle degenerative symptoms, (rather than state that it can).

REVIEWER COMMENTS

AUTHORS' POINT-BY-POINT RESPONSE

Reviewer #1 (Remarks to the Author):

The manuscript by Dennhag and colleagues is focused on the role of *fhl2b* as having ameliorative properties in the zebrafish dystrophin-deficient model of Duchenne muscular dystrophy (DMD). Extraocular muscles (EOM) in DMD are spared from muscle wasting and dystrophic symptoms in patients. The authors compared differences in gene expression as a response to muscle dystrophies between the EOMs and trunk muscle in zebrafish via transcriptomic profiling. The authors demonstrate that *Fhl2* is upregulated in response to knockout of desmin, plectin and obscurin, intermediate filament proteins whose knockout causes different muscle dystrophies, and contributes to DMD disease protection. *Fhl2b* ectopic overexpression can partially rescue dystrophic symptoms in the *sapje*/DMD model. The authors conclude that FHL2 overexpression can protect against dystrophic pathologies.

This is a very interesting study that integrates several model systems in zebrafish to identify a protective mechanism against muscular dystrophy that would be of high interest to the muscular dystrophy and general muscle biology fields. However, some key questions with regards to the extrapolation of findings from that of the extraocular muscles to that of *fhl2b* transgenic overexpression in dystrophic/dmd mutant zebrafish remain. Questions with regards to the mechanism of protection and FHL2's function need to be elucidated. Methodology is appropriate and statistical tests are appropriately used. The statistical approach and design are appropriate. This manuscript is well-written and with some elucidation of key experiments, would be a valuable addition to the muscular dystrophy field.

We thank reviewer #1 for the constructive comments and the introductory remarks. We have performed new experiments and added discussion topics based on the suggestions. Overall, this has led to a significant improvement of the study. Please see the point-by-point responses below. Please note that the line numbers refer to the manuscript version with tracked changes.

Overall Comments:

1. There's some overlapping expression and suggestive overlapping function between the FHL proteins. Does FHL1 and/or FHL3 increase in expression in the *fhl2a*, *fhl2b* and/or double *fhl2a/b* knockout zebrafish?

To assess this, we analyzed the expression of *fhl1a*, *fhl1b*, *fhl3a* and *fhl3b* in WT, *desma*^{-/-}; *desmb*^{-/-}, *desma*^{-/-}; *desmb*^{-/-}; *fhl2a*^{-/-}, *desma*^{-/-}; *desmb*^{-/-}; *fhl2b*^{-/-} and *desma*^{-/-}; *desmb*^{-/-}; *fhl2a*^{-/-}; *fhl2b*^{-/-} mutants by qPCR. None of these analyses indicated a compensatory mechanism by increased expression of *fhl1a*, *fhl1b*, *fhl3a* or *fhl3b*. We did however see a reduced expression of both *fhl1* paralogs in the *desma*^{-/-}; *desmb*^{-/-}; *fhl2a*^{-/-}; *fhl2b*^{-/-} mutants. This data is included in the results section (lines 263-271) of the revised manuscript, in New Fig. S4 f-1 and included in the discussion (lines 571-574).

2. The authors mention the lack of publications demonstrating FHL2 is expressed extraocular muscles. Have the authors compared the transcriptomes from the *fhl2b* EOMs fish to that of published mouse studies demonstrating that EOM's are resistant to dystrophic pathologies (Pacheco-Pinedo et al., *Physiol. Genom.*, 2009)?

It is correct that there are some published transcriptomic studies of extraocular muscles (Niemann CU et al., *Neurol Sci*, 2000., Porter JD et al., *Proc Natl Acad Sci USA*, 2001, Gorospe JR et al., *Physiol. Genom.*, 2002, Fischer MD et al., *Physiol. Genom.*, 2005, Pacheco-Pinedo et al., *Physiol. Genom.*, 2009, Valentina Taglietti et al., *Sci Transl Med*, 2023). However, these studies include only healthy test groups and focus on the difference between EOMs and body muscle or progenitor cells alone. Fhl2 is not identified in Pacheco-Pinedo et al., *Physiol. Genom.*, 2009, however, this study is focused on the "side population" progenitor cell population of the EOMs and TAs and does not contain the other cell types, which may explain the lack of Fhl2. Therefore, we did not include these results in the discussion.

3. Line 204 "however, *fhl2b* likely has an additional role in hypertrophic protection". How exactly do the authors know that Fhl2b has an additional hyperprotective role? Can the authors explain why Fhl2a overexpression was not evaluated?

The reasoning behind the statement in line 204 in the original submission is based on the significant increase in myofiber area in EOMs lacking *fhl2b* (Fig. 3A-B). In contrast, mutation of *fhl2a* (in addition to the *desma* and *desmb* genes) did not result in any changes of the myofiber area. Based on these results we concluded *fhl2b* to be primary responsible for the resulting signs of hypertrophy in the quadruple mutants. We have now tried to clarify this and the sentence in the revised manuscript (lines 299-301) reads: "...however, *fhl2b* likely has an additional role in hypertrophic protection given the increase in myofiber area observed in the examined mutants lacking *fhl2b* (Fig. 3a-b)". Additionally, the first sentence in the following paragraph (lines 305-306) now reads "Since *fhl2b* was the only Fhl2 paralog differentially expressed in our transcriptomic data of EOMs lacking *desmin*, we chose to investigate whether *fhl2b* also could protect muscles other than EOMs in muscular dystrophy." To clarify the reasoning behind only analyzing *fhl2b* as opposed to both *fhl2* genes.

4. FHL2 has been shown to modify the beta-adrenergic stimulation response in hypertrophic hearts in mice (Kong et al., *Circulation*, 2001). The authors demonstrate here that Fhl2 causes EOM myofiber hypertrophy (Figure 3). Did the authors evaluate hypertrophy in the heart and/or skeletal muscle of their *fhl2b* and/or *fhl2a/b* double KO mutants?

In order to examine hypertrophy in cardiac muscle and skeletal muscle, we performed new experiments and statistical analyses. We examined the histology and size of the ventricle in 12 months old WT, *desma*^{-/-};*desmb*^{-/-}, *desma*^{-/-};*desmb*^{-/-};*fhl2a*^{-/-}, *desma*^{-/-};*desmb*^{-/-};*fhl2b*^{-/-} and *desma*^{-/-};*desmb*^{-/-};*fhl2a*^{-/-};*fhl2b*^{-/-} mutants in correlation to body size, but did not find any significant differences between the test groups (New Fig S5). Similar examinations and quantifications were made in cross sections of trunk skeletal musculature, where none of the compared slow- and fast specific myofiber areas among the examined mutants indicated a significant difference between test groups (New Fig. S5). These results are included in the

results section (lines 294-297) and discussed in relation to other results (lines 551-554) in the revised manuscript.

5. Similar to my previous point, FHL2 has been shown to prevent pathological cardiac, and likely other tissue, growth (Hojayav et al., Mol. Cell. Biol. 2012) via a calcineurin-dependent pathway. Did the authors identify any calcineurin pathways that were affected in the overexpressed transgenic fish?

There are indeed a number of published papers that link Fhl2 to WNT, β -Catenin and Calcineurin-signaling. We agree that this may have been overlooked in our original submission and are thankful to reviewer 1 for pointing this out. We did not find any significant upregulation of Calcineurin in our analysis of *fhl2b* overexpressing zebrafish and the WNT-signaling pathway as a whole was not significantly altered in the gene ontology analysis (14/902 DEGs). In addition, our transcriptomic analysis of EOMs did not show dysregulation of Calcineurin. To expand our discussion regarding the WNT-signaling pathway, we performed new qPCR experiments where we confirmed numerous members in the WNT-signaling context to be significantly affected by mutation of the *fhl2* paralogs, including β -Catenin. This data is now included in the results section (New Fig. S4j-n, lines 271-277) and discussed in relation to previously published material, including Hojayav et al., 2012 (lines 543-551).

6. The biggest claim of innervation in response to Fhl2b transgenic overexpression needs a bit more follow-up. The authors show some bungarotoxin (Sup. Fig. S7; Figure 5E-G) staining but no quantification of this or demonstration of functional improvement as a result of increased innervation. The authors should really quantify the percentage of innervation in response to Fhl2b overexpression in the *dmd* mutants, and perhaps perform some additional acetylcholinesterase assessments of the transgenic muscles.

We performed new experiments and analyses to quantify innervation, which are included in New Fig. 5 of the revised manuscript. To examine and measure NMJs, we calculated and compared the amount of α -BTX+ NMJs in WT, *dmd*^{-/-} and *dmd*^{-/-}; *Tg(503unc:fhl2b-T2A-EGFP)* using the Imaris software and found that the overexpression of *fhl2b* significantly improved the NMJs compared to *dmd*^{-/-} (New Fig. 5k). We also found that the overlap between α -BTX and SV2 in a Z-stack was significantly improved in *dmd*^{-/-}; *Tg(503unc:fhl2b-T2A-EGFP)* compared to *dmd*^{-/-} (New Fig. 5l), indicating increased innervation. We also analyzed and quantified the total motor axon lengths and volumes within trunk muscles of WT, *dmd*^{-/-} and *dmd*^{-/-}; *Tg(503unc:fhl2b-T2A-EGFP)* larvae using the Imaris software and found that the overexpression of *fhl2b* significantly rescued the *dmd*^{-/-} phenotype (New Fig. 5m-p). The revised data is included in the result section (lines 413-426).

7. Another interesting finding, or lack thereof, by the authors is that there appears to be little affect on WNT signaling via Fhl2 overexpression if I am interpreting the transcriptomic data correctly? Did the authors directly probe WNT signaling in either the transgenic *fhl2b* fish or knockout fish? FHL2 has been shown to be a direct interactor with beta-catenin to regulate WNT signaling and muscle differentiation. I would be curious to know if this is or is not the case in the authors' model(s).

It is correct that the overexpression of *fhl2b* in siblings does not cause a significant impact on WNT-signaling. However, as stated under comment number 5 above, we agree that this is an important question to address, so we have analyzed the expression of a number of genes in the WNT/ β -Catenin pathway in WT, *desma*^{-/-};*desmb*^{-/-}, *desma*^{-/-};*desmb*^{-/-};*fhl2a*^{-/-}, *desma*^{-/-};*desmb*^{-/-};*fhl2b*^{-/-} and *desma*^{-/-};*desmb*^{-/-};*fhl2a*^{-/-};*fhl2b*^{-/-} mutants and have confirmed that both *fhl2a* and *fhl2b* respectively are involved in its regulation. We have added this data in the results section (New Fig. S4j-n, lines 271-277) and a discussion in relation to previously published material (lines 543-554).

Reviewer #2 (Remarks to the Author):

This manuscript first describes a new zebrafish dystrophy model followed by an investigation of the transcriptomics of extraocular versus trunk muscle (EOM being usually spared in dystrophy). This revealed that the four and a half LIM domain protein *fhl2* is upregulated in EOM in three different zebrafish dystrophies and suggests it may lead to the protection in EOM. Overexpression of *fhl2b* in the most severe zebrafish dystrophy model improved their survival, suggesting *Fhl2* as a potential target for treatment.

This research is extremely interesting, since members of the FHL family (especially FHL1 and FHL2) were shown previously to be involved in the stress response of skeletal muscle and heart during disease. Previous studies have demonstrated that at least in mouse FHL2 is NOT expressed in skeletal muscle (Lange et al., 2002) and only expressed in the heart and FHL3 was reckoned to be the main FHL in skeletal muscle (Fimia et al., 2000; Morgan and Madgwick, 1999). However, EOM are notorious for their peculiar status a bit inbetween cardiac and skeletal muscle, so the FHL2 expression in them may well also be the case in mouse/human - this should be shown by Western blot to provide more support for the potential translational value of this study. Unfortunately the images in Figure 2G and H that show FHL2 expression by immunofluorescence in human and mouse EOM are of too low quality to conclude much (signal in G extremely weak, no counterstain of any sort; why is the signal in the mouse in H suddenly only peripheral in the fibres when it appeared to be throughout the cytoplasm in fish and potentially human?). In addition, the expression levels of the zebrafish homologues of *Fhl3* should be shown to account for upregulation/downregulation in the different experimental scenarios.

We thank reviewer #2 for the constructive comments in the introductory remarks, they are much appreciated. To improve the quality of Fig. 2G and H in the previous submission, we performed new IHC experiments, including laminin and DAPI as counter stains, which significantly improved the readability and interpretation of the data. The new panels are included in the results section (lines 233-235) under New Fig. 2g and i. Please note that the line numbers refer to the manuscript version with tracked changes.

Specific comments:

1. Show *Fhl2* expression in mouse and human EOM by Western blot.

We have performed western blots on human and mouse EOMs, which confirm the presence of FHL2 in these tissues. These results are presented in conjunction with the corresponding IHC panels in New Fig. 2 g-j (lines 233-235).

2. Analyse fhl3 expression in the various models

We have now analyzed the expression of *fhl1a*, *fhl1b*, *fhl3a* and *fhl3b* in WT, *desma*^{-/-};*desmb*^{-/-}, *desma*^{-/-};*desmb*^{-/-};*fhl2a*^{-/-}, *desma*^{-/-};*desmb*^{-/-};*fhl2b*^{-/-} and *desma*^{-/-};*desmb*^{-/-};*fhl2a*^{-/-};*fhl2b*^{-/-} mutants by qPCR. This data is included in the results section (lines 263-271), discussion (lines 571-574) and in New Fig. S4f-i of the revised manuscript.

3. FHL2 was shown in other system to be potentially involved in gene transcription. While the authors report that they did not see any signal for Fhl2 in the nucleus, its residence time there might be really brief and only detectable by inhibition of nuclear export pathways. The observation that it is actually the communication between muscle cells and nerves that is improved in the “rescue” fish is intriguing and I think necessitates a targeted investigation whether the Fhl2b fish have higher expression levels of miR-206 - which is well known to mediate communication between these two cell types.

To further investigate the role of Fhl2 in the communication between nerve and muscle, we performed several new experiments including analysis of *mir-206*. In short, overlap between α -BTX and SV2 in a Z-stack and the total neuronal lengths and volumes of neurons was significantly improved in *dmd*^{-/-}; *Tg(503unc:fhl2b-T2A-EGFP)* compared to *dmd*^{-/-}, which is included in the revised manuscript in New Fig. 5j-p (lines 413-426). The expression of *mir-206* was significantly elevated in *dmd*^{-/-} compared to sibling controls, which is indicative of a high degree of ongoing muscle/nerve regeneration in the *dmd*^{-/-} due to the severe muscle damage in these larvae, and is in line with previous literature (Cacciarelli et al., 2011, EMBO mol med; Liu et al., 2012, J Clin Invest). *mir-206* was also significantly elevated in *dmd*^{-/-}; *Tg(503unc:fhl2b-T2A-EGFP)*, but to a lesser degree as compared to *dmd*^{-/-} larvae, which correlates with a lesser degree of muscle damage and an overall improved cellular integrity. This data is included in the revised manuscript (New Fig. 5e, lines 385-392). Furthermore, the expression of *hdac4*, suggested to be regulated by *mir-206* and also a negative biomarker for DMD followed the same pattern as *mir-206* (New Fig. 5f, lines 385-392).

4. Discuss Okamoto et al., 2012 FASEB & Friedrich et al., 2014; Basic Res Cardiol (FHL2 overexpression in cardiomyopathy); Hojayeve et al., 2012 (FHL2 & calcineurin)

We thank reviewer #2 for the suggested references, which we have discussed and included in the revised manuscript in the second paragraph of the discussion (Lines 519-554).

Reviewer #3 (Remarks to the Author):

This manuscript examines the role of the gene *fhl2b* for its potential role in sparing the extraocular muscles in various forms of muscular dystrophy using a large number of zebrafish genetic mutants. This study is very well done, and the results are very exciting. The improved function of the body skeletal muscle after up-regulated expression of *fhl2b* is quite striking, and the data are important. I have only a few suggestions to improve readability

and clarity, and all are relatively minor and straightforward. I also have made suggestions of literature that should be included in the Discussion section.

We thank reviewer #3 for the constructive comments regarding the formatting of the manuscript and figures. Overall, we have tried to incorporate as many of the suggestions as possible to increase readability and quality, which has improved the quality of the study. Please see the point-by-point response below. Please note that the line numbers refer to the manuscript version with tracked changes.

1. Please include supplemental figure 1 as a regular figure as well as supplemental figure 9. There is basic information in these figures, as they relate to the entire study, within these figures.

The figures in the revised manuscript have been rearranged in an attempt to improve the manuscript according to the constructive suggestions of reviewer #3. The previous supplemental Fig. 1 (G-L) has been incorporated into New Fig. 1 (panels k-r) of the revised manuscript. Due to space issues, the descriptive part, and the control of the generated desmin mutants had to be kept as supplemental material in New Fig. S1. The original submission did not include a supplemental Fig. 9, which is referred to in the above comment. We assumed that this comment referred to supplemental Fig. 8 and have therefore included more data from supplemental Fig. S8 into New Fig. 7, specifically the neutrophil analysis with statistics (New Fig. 7c, d) and rearranged several of the supplemental figures and hope that this can be considered as an overall improvement of the organization of the manuscript.

2. There are a number of studies in adult mammalian skeletal muscle showing that FHL2 is shown to be expressed in myogenic progenitor cells, and this literature should be discussed. FHL2 was shown to interact with FOXK1 and promoted proliferation of myogenic progenitor cell populations (Martin et al., J Cell Biol. 2002; Shi et al. Stem Cells 2010). This was shown in other studies of skeletal muscle satellite cells (Zhu et al., Genes 2022). Shi et al. (2010) also show that fhl2 null mice have perturbed muscle regeneration specifically through cell cycle arrest, and that FHL2 expression correlated with expression of MyoD and Myogenin. These various results and what they mean in the context of your very interesting study need to be added to the Discussion, particularly where attributing other roles and locations for this protein (e.g. in the paragraph starting on line 331).

We are very thankful for the constructive suggestion and have expanded our discussion regarding the potential role of Fhl2 in muscle regeneration in the revised manuscript, where the above-mentioned literature now is cited. We have also expanded the discussion to cover additional potential roles of Fhl2, as suggested (lines 519-543, 590-603). We hope that this will provide the readers with a more comprehensive analysis of our data in light of previous publications.

3. In addition, while you did not detect FHL2B-positive myonuclei in the EOMs, it may well be that precursor cells express it. This needs to be discussed, particularly in light of the Pax7-positive precursor cells you show that others were unable to demonstrate within the EOM of zebrafish (Saera-Vila et al., IOVS. 2015). It is hard to tell in Figure 6, for example, where the

Pax7-expressing cells are located. In the controls, they seem to be within the connective tissue rather than associated with muscle fibers. While removal of debris is an important factor here, it is likely (based on the literature) that the muscle precursor cells are more active in the presence of elevated levels of FHL2B and more numerous in the EOMs.

We show that Pax7 positive cells are present in the skeletal muscle and accumulated at injury sites within the zebrafish trunk in Fig. 6 in the original submission. The location of Pax7 cells in the trunk skeletal muscle of zebrafish larvae is well documented in literature. We do however not show or claim Pax7 to be present in the EOMs. We have re-arranged the original Fig. 6 (New Fig. 7 in the revised manuscript) and in an attempt to clarify this and improve readability, we separated the channels for Pax7 and F-actin (Fig. 7 g, i). We have also included the Mpx analysis of the injury site (New Fig. 7c) and have moved the time-lapse analysis (previous Fig. 6A-C) to New Fig. 6c-e in the revised manuscript. We have also enlarged the size and improved the quality of the illustrative cartoons to further help the reader. We hope that these changes will increase the readability of the manuscript.

4. There is also evidence that FHL2 interacts with muscle integrin receptors, which would play a role in mechanical stabilization of the muscle cells (Samson et al., 2004). This potential sparing mechanism also should be discussed in the Discussion section.

We agree with reviewer #3 that this is an important point and have cited, discussed and made proposals regarding the literature in the revised manuscript in the discussion (lines 590-603) regarding the potential role of Fhl2b in mechanical stabilization of the sarcolemma.

5. Because you are comparing multiple groups in almost all your graphs, a box plot gives a cleaner, more easily understandable summary of the data. The median is clearer. I suggest using box plots for displaying data summaries. They also will show clear outliers in the data.

We are following the journal guidelines regarding font size and type of graphs. We have chosen to use violin plots to better be able to illustrate individual measure points and hope that this is acceptable.

6. The figures are rather small. The text is hard to read, even when I am reading this on my giant computer monitor. I suggest reducing the white space and enlarging the figures so that the details of the histology are clearer, as well as for the graphs, so that the labels can all be read.

We have tried to maximize the panel sizes and can unfortunately not increase them more without significantly increasing the number of figures in the manuscript. We have tried to reduce the white space as much as possible and have tried to improve readability throughout the manuscript by including new and improved illustrative cartoons, arrows and we have increased magnifications where relevant, following the instructions of the journal.

7. Figure 5G is out of focus; please replace these photos. In 5G control, are each of the yellow dots a separate postsynaptic part of the neuromuscular junction? Are the dark spaces all single muscle fibers in cross section? It is quite difficult to assess what one is seeing. A

crisper image and increased size will certainly help, but it might be useful to provide a photograph of a bright field image.

The panels in Fig. 5 are merged confocal Z-stacks of whole-mount zebrafish larvae. To illustrate the region analyzed in the panels of New Fig. 5 we have included an illustrative cartoon in the revised manuscript to better show the analyzed region and to guide the reader. In the revised New Fig. 5, we have added new data (New Fig. 5i-p) and have changed panels for improved quality (New Fig. 5i). Overall, we have reorganized figures to include more data, both from new experiments and from previous supplemental material (in response to comments by other reviewers) in the revised manuscript. In addition, we have analyzed the IHC Z-stacks in New Fig. 5 with Imaris software to better illustrate NMJs by α -BTX and motor axons by acetylated tubulin and have included quantifications to support our observations.

Other

8. Figure title for Figure 2 would be more accurate as “fhl2b is upregulated in the EOM in response to desmin-related muscular dystrophy”.

We thank reviewer #3 for this suggestion and have now changed the title for New Fig. 2 accordingly.

9. It would be good to temper line 329 to say that fhl2b therapy has the potential to alleviate muscle degenerative symptoms, (rather than state that it can).

We thank reviewer #3 for this suggestion and have now changed it from “can” to “has the potential to” (line 517).

We hope that the manuscript can now be accepted for publication in Nature Communications.

Yours sincerely,

Fatima Pedrosa Domellöf and Jonas von Hofsten

REVIEWERS' COMMENTS

Reviewer #1 (Remarks to the Author):

The authors have made extensive clarifications and revisions in the resubmitted version of this manuscript. The noteworthy findings are with regards to the overexpression of fh12b in the EOM muscles as protective in dystrophic muscle. These findings and the others are of interest to the DMD biology and general muscle biology fields. I appreciate their clarification on why fh12b was pursued in analysis of hypertrophic studies over fh12a. Their cardiac analysis is also quite convincing in many regards. Statistical tests are appropriately used and the methodology is sound. Sample sizes for experiments are appropriate. The work is overall supportive of the conclusions drawn by the authors. I have no additional concerns.

Reviewer #2 (Remarks to the Author):

My concerns were all addressed and the new data have made the manuscript even more compelling. Just a very minor thing remains to be done, add to Figure legend S4f that the expression analysis was done by qPCR.

Reviewer #3 (Remarks to the Author):

The authors have significantly modified this submission. They have deal with all the concerns I had raised.

I think it is exciting and important work.

Point-by-point response to the reviewers' comments

REVIEWERS' COMMENTS

Reviewer #1 (Remarks to the Author):

The authors have made extensive clarifications and revisions in the resubmitted version of this manuscript. The noteworthy findings are with regards to the overexpression of fh12b in the EOM muscles as protective in dystrophic muscle. These findings and the others are of interest to the DMD biology and general muscle biology fields. I appreciate their clarification on why fh12b was pursued in analysis of hypertrophic studies over fh12a. Their cardiac analysis is also quite convincing in many regards. Statistical tests are appropriately used and the methodology is sound. Sample sizes for experiments are appropriate. The work is overall supportive of the conclusions drawn by the authors. I have no additional concerns.

Response: We are pleased to read that reviewer #1 is satisfied with the previous revision of our manuscript.

Reviewer #2 (Remarks to the Author):

My concerns were all addressed and the new data have made the manuscript even more compelling. Just a very minor thing remains to be done, add to Figure legend S4f that the expression analysis was done by qPCR.

Response: We are thankful for the comment by reviewer #2 and have included the clarification regarding the use of qPCR in the figure legend of Fig. S4 of the revised version of our manuscript.

Reviewer #3 (Remarks to the Author):

The authors have significantly modified this submission. They have deal with all the concerns I had raised.

I think it is exciting and important work.

Response: We are happy to read that reviewer #3 is satisfied with the previous revision of our manuscript.